

# Indicators of Global Climate Change 2023: annual update of key indicators of the state of the climate system and human influence

Piers M. Forster[1], Chris Smith[1,2,3], Tristram Walsh[4], William F. Lamb[5,1], Robin Lamboll[6], Bradley Hall[23], Mathias Hauser[7], Aurélien Ribes[8], Debbie Rosen[1], Nathan P. Gillett[9], Matthew D. Palmer[3,10], Joeri Rogelj[6], Karina von Schuckmann[11], Blair Trewin[12], Myles Allen[4], Robbie Andrew[13], Richard A. Betts[3,18], Tim Boyer[15], Carlo Buontempo[14], Samantha Burgess[14], Chiara Cagnazzo[14], Lijing Cheng[16], Pierre Friedlingstein[18,19], Andrew Gettelman[38], Johannes Gütschow[20], Masayoshi Ishii[22], Stuart Jenkins[4], Xin Lan[21,35], Colin Morice[3], Jens Mühle[42], Christopher Kadow[23], John Kennedy[24], Rachel E. Killick[3], Paul B. Krummel[41], Jan C. Minx[5,1], Gunnar Myhre[13], Vaishali Naik[17], Glen P. Peters[13], Anna Pirani[25], Julia Pongratz[26,34], Carl-Friedrich Schleussner[27], Sonia I. Seneviratne[7], Sophie Szopa[28], Peter Thorne[29], Mahesh V. M. Kovilakam[38], Elisa Majamäki[39], Jukka-Pekka Jalkanen[39], Margreet van Marle[40], Rachel M. Hoesly[37], Robert Rohde[30], Dominik Schumacher[7], Guido van der Werf[36], Russell Vose[31], Kirsten Zickfeld[32], Xuebin Zhang[9], Valerie Masson-Delmotte[28], Panmao Zhai[33]

[1]Priestley Centre, University of Leeds, Leeds, LS2 9JT, UK
[2]International Institute for Applied Systems Analysis (IIASA), Austria
[3]Met Office Hadley Centre, Exeter, UK
[4]Environmental Change Institute, University of Oxford, UK
[5]Mercator Research Institute on Global Commons and Climate Change (MCC), Berlin, Germany
[6]Centre for Environmental Policy, Imperial College London, UK
[7]Institute for Atmospheric and Climate Science, Department of Environmental Systems Science, ETH Zurich, Zurich, Switzerland
[8]Université de Toulouse, Météo France, CNRS, France
[9]Environment and Climate Change Canada, Canada
[10]School of Earth Sciences, University of Bristol, UK
[11]Mercator Ocean international, Toulouse, France
[12]Bureau of Meteorology, Melbourne, Australia
[13]CICERO Center for International Climate Research, Oslo, Norway
[14]ECMWF, Bonn, Germany
[15]NOAA's National Centers for Environmental Information (NCEI), Silver Spring, MD, USA
[16]Institute of Atmospheric Physics, Chinese Academy of Sciences, Beijing, China
[17]NOAA Geophysical Fluid Dynamics Laboratory, Princeton, NJ, USA
[18]Faculty of Environment, Science and Economy, University of Exeter, UK
[19]Laboratoire de Météorologie Dynamique/Institut Pierre-Simon Laplace, CNRS, Ecole Normale Supérieure/Université PSL, Paris, France





[20]Climate Resource, Australia/Germany
[21]NOAA Global Monitoring Laboratory, Boulder, CO, USA
[22]Meteorological Research Institute, Tsukuba, Japan
[23]German Climate Computing Center, Hamburg, Germany (DKRZ)
[24]No affiliation, Verdun, France.
[25]Euro-Mediterranean Center on Climate Change (CMCC), Venice, Italy; Università Cà Foscari,Venice,  Italy
[26]Ludwig-Maximilians-Universität München, Germany
[27]Climate Analytics, Berlin, Germany and Geography Department and IRI THESys, Humboldt-Universität zu Berlin,
Berlin, Germany
[28]Institut Pierre Simon Laplace, Laboratoire des sciences du climat et de l'environnement, UMR8212 CNRS-CEA-
UVSQ, Université Paris-Saclay, 91191, Gif-sur-Yvette, France
[29]ICARUS Climate Research Centre, Maynooth University, Maynooth, Ireland
[30]Berkeley Earth, Berkeley, CA, USA
[31]NOAA's National Centers for Environmental Information (NCEI), Asheville, NC, USA
[32]Simon Fraser University, Vancouver, Canada
[33]Chinese Academy of Meteorological Sciences, Beijing, China
[34]Max Planck Institute for Meteorology, Hamburg, Germany
[35]CIRES, University of Colorado Boulder, Boulder, CO, USA
[36]Wageningen University and Research, Wageningen, The Netherlands
[37]Pacific Northwest National Laboratory, Richland, WA, USA
[38]LARC,NASA, USA
[39]Finnish Meteorological Institute, Helsinki, Finland
[40]Delteras, Delft, The Netherlands
[41]Global Systems Institute, University of Exeter, UK
[42]Scripps Institution of Oceanography, University of California San Diego, La Jolla, CA, USA
*Correspondence to*: Piers. M. Forster (p.m.forster@leeds.ac.uk)

**Abstract.**
Intergovernmental Panel on Climate Change (IPCC) assessments are the trusted source of scientific evidence for
climate negotiations taking place under the United Nations Framework Convention on Climate Change (UNFCCC).
Evidence-based decision-making needs to be informed by up-to-date and timely information on key indicators of the
state of the climate system and of the human influence on the global climate system. However, successive IPCC
reports are published at intervals of 5–10 years, creating potential for an information gap between report cycles.

We follow methods as close as possible to those used in the IPCC Sixth Assessment Report (AR6) Working Group
One (WGI) report. We compile monitoring datasets to produce estimates for key climate indicators related to forcing
of the climate system: emissions of greenhouse gases and short-lived climate forcers, greenhouse gas concentrations,
radiative forcing, the Earth's energy imbalance, surface temperature changes, warming attributed to human activities,
the remaining carbon budget, and estimates of global temperature extremes. The purpose of this effort, grounded in
an open data, open science approach, is to make annually updated reliable global climate indicators available in the
public domain (https://doi.org/10.5281/zenodo.11064126 Smith et al., 2024a). As they are traceable to IPCC report



methods, they can be trusted by all parties involved in UNFCCC negotiations and help convey wider understanding
of the latest knowledge of the climate system and its direction of travel.

The indicators show that, for the 2014–2023 decade average, observed warming was 1.19 [1.06 to 1.30] °C, of which
1.19 [1.0 to 1.4] °C was human-induced. For the single year average, human-induced warming reached 1.31 [1.1 to
1.7] °C in 2023 relative to 1850-1900. This is below the 2023 observed record of 1.43 [1.32 to 1.53] °C, indicating a
substantial contribution of internal variability in the 2023 record. Human-induced warming has been increasing at rate
that is unprecedented in the instrumental record, reaching 0.26 [0.2 - 0.4] °C per decade over 2014-2023. This high
rate of warming is caused by a combination of greenhouse gas emissions being at an all-time high of $54 \pm 5.4$ GtCO2e
per year over the last decade, as well as reductions in the strength of aerosol cooling. Despite this, there is evidence
that the rate of increase in $CO_2$ emissions over the last decade has slowed compared to the 2000s, and depending on
societal choices, a continued series of these annual updates over the critical 2020s decade could track a change of
direction for some of the indicators presented here.
**1 Introduction**
The IPCC AR6 has provided an assessment of human influence on key indicators of the state of climate grounded in
data up to year 2019 (IPCC WGI 2021, Supplement Sect. S1). The next IPCC AR7 assessment report is due towards
the end of the decade. Given the speed of recent change, and the need for updated climate knowledge to inform
evidence-based decision-making, the Indicators of Global Climate Change (IGCC) was initiated to provide
policymakers with annual updates of the latest scientific understanding on the state of selected critical indicators of
the climate system and of human influence.
This second annual update follows broadly the format of last year (Forster et al., 2023), focussing on indicators related
to heating of the climate system, building from greenhouse gas emissions towards estimates of human-induced
warming and the remaining carbon budget. Fig. 1 presents an overview of the aspects assessed and their interlinkages
from cause (emissions) through effect (changes in physical indicators) to Climatic Impact-Drivers. It also provides a
visual roadmap as to the structure of remaining sections in this paper to guide the reader.










**Figure 1 The flow chart of data production from emissions to human induced warming and the remaining carbon budget, illustrating both the rationale and workflow within the paper production.**

The update is based on methodologies assessed by the IPCC Sixth Assessment Report (AR6) of the physical science basis of climate change (Working Group One (WGI) report; IPCC, 2021a) as well as Chap. 2 of the WGIII report (Dhakal et al., 2022) and is aligned with the efforts initiated in AR6 to implement FAIR (Findable, Accessible, Interoperable, Reusable) principles for reproducibility and reusability (Pirani et al., 2022; Iturbide et al., 2022). IPCC reports make a much wider assessment of the science and methodologies – we do not attempt to reproduce the comprehensive nature of these IPCC assessments here. Our aim is to rigorously track both climate system change and methodological improvements between IPCC report cycles, thereby transparency and consistency in between successive reports.

The update is organised as follows: emissions (Sect. 2) and greenhouse gas (GHG) concentrations (Sect. 3) are used to develop updated estimates of effective radiative forcing (Sect. 4). Earth's energy imbalance (Sect. 5) and observations of global surface temperature change (Sect. 6) are key global indicators of a warming world. The contributions to global surface temperature change from human and natural influences are formally attributed in Sect. 7, which tracks the level and rate of human-induced warming. Sect. 8 updates the remaining carbon budget to policy-relevant temperature thresholds. Sect. 9 gives an example of global-scale indicators associated with climate extremes of maximum land surface temperatures. An important purpose of the exercise is to make these indicators widely available and understood. Code and data availability are given in Sect. 10, and conclusions are presented in Sect. 11. Data are available at https://zenodo.org/records/11064126 (Smith et al., 2024a).

## 2 Emissions

Historic emissions from human activity were assessed in both AR6 WGI and WGIII. Chapter 5 of WGI assessed $CO_2$ and $CH_4$ emissions in the context of the carbon cycle (Canadell et al., 2021). Chapter 6 of WGI assessed emissions in the context of understanding the climate and air quality impacts of short-lived climate forcers (Szopa et al., 2021). Chapter 2 of WGIII, published one year later (Dhakal et al., 2022), assessed the sectoral sources of emissions and gave the most up-to-date understanding of the current level of emissions. This section bases its methods and data on those employed in this WGIII chapter.

### 2.1 Methods of estimating greenhouse gas emissions changes

Like in AR6 WGIII, net GHG emissions in this paper refer to releases of GHGs from anthropogenic sources minus removals by anthropogenic sinks, for greenhouse gases reported under the common reporting format of the UNFCCC. This includes $CO_2$ emissions from fossil fuels and industry ($CO_2$-FFI); net $CO_2$ emissions from land use, land-use change and forestry ($CO_2$-LULUCF); $CH_4$; $N_2O$; and fluorinated gas (F-gas) emissions. $CO_2$-FFI mainly comprises fossil-fuel combustion emissions, as well as emissions from industrial processes such as cement production. This excludes biomass and biofuel use. $CO_2$-LULUCF is mainly driven by deforestation but also includes anthropogenic



removals on land from afforestation and reforestation, emissions from logging and forest degradation, and emissions
and removals in shifting cultivation cycles, as well as emissions and removals from other land-use change and land
management activities, including peat burning and drainage. The non-$CO_2$ GHGs – $CH_4$, $N_2O$ and F-gas emissions –
are linked to the fossil-fuel extraction, agriculture, industry and waste sectors.

Global regulatory conventions have led to a twofold categorisation of F-gas emissions (also known as halogenated
gases). Under UNFCCC accounting, countries record emissions of hydrofluorocarbons (HFCs), perfluorocarbons
(PFCs), sulfur hexafluoride ($SF_6$) and nitrogen trifluoride ($NF_3$) – hereinafter "UNFCCC F-gases". However, national
inventories tend to exclude halons, chlorofluorocarbons (CFCs) and hydrochlorofluorocarbons (HCFCs) – hereinafter
"ODS (ozone-depleting substance) F-gases" – as they have been initially regulated under the Montreal Protocol and
its amendments. In line with the WGIII assessment, ODS F-gases and other substances, are not included in our GHG
emissions reporting but are included in subsequent assessments of concentration change (including compounds formed
in the atmosphere as ozone), effective radiative forcing, human-induced warming, carbon budgets and climate impacts
in line with the WGI assessment.

There are also varying conventions used to quantify $CO_2$-LULUCF fluxes. These include the use of bookkeeping
models, dynamic global vegetation models (DGVMs) and aggregated national inventory reporting (Pongratz et al.,
2021). Each differs in terms of their applied system boundaries and definitions and they are not directly comparable.
However, efforts to "translate" between bookkeeping estimates and national inventories using DGVMs have
demonstrated a degree of consistency between the varying approaches (Friedlingstein et al., 2022; Grassi et al., 2023).

Each category of GHG emissions included here is covered by varying primary sources and datasets. Although many
datasets cover individual categories, few extend across multiple categories, and only a minority have frequent and
timely update schedules. The Global Carbon Budget (GCB; Friedlingstein et al., 2023) covers $CO_2$-FFI and $CO_2$-
LULUCF. The Emissions Database for Global Atmospheric Research (EDGAR; Crippa et al., 2023) and the Potsdam
Real-time Integrated Model for probabilistic Assessment of emissions Paths (PRIMAP-hist; Gütschow et al., 2016;
Gütschow et al., 2024) cover $CO_2$-FFI, $CH_4$, $N_2O$ and UNFCCC F-gases. The Community Emissions Data System
(CEDS; Hoesly et al. 2018; ; Hoesly and Smith, 2024) covers $CO_2$-FFI, $CH_4$, and $N_2O$. The Global Fire Emissions
Database (GFED; van der Werf et al., 2017) covers $CO_2$, $CH_4$, and $N_2O$. As detailed below, for various reasons not
all these datasets were employed in this update.

In AR6 WGIII, total net GHG emissions were calculated as the sum of $CO_2$-FFI, $CH_4$, $N_2O$ and UNFCCC F-gases
from EDGAR, and net $CO_2$-LULUCF emissions from the GCB. Net $CO_2$-LULUCF emissions followed the GCB
convention and were derived from the average of three bookkeeping models (Hansis et al., 2015; Houghton and
Nassikas, 2017; Gasser et al., 2020). Version 6 of EDGAR was used (with a fast-track methodology applied for the
final year of data – 2019), alongside the 2020 version of the GCB (Friedlingstein et al., 2020). $CO_2$-equivalent
emissions were calculated using global warming potentials with a 100-year time horizon (GWP100 henceforth) from



AR6 WGI Chap. 7 (Forster et al., 2021). Uncertainty ranges were based on a comparative assessment of available data
and expert judgment, corresponding to a 90 % confidence interval (Minx et al., 2021): ±8 % for $CO_2$-FFI, ±70 % for
$CO_2$-LULUCF, ±30 % for $CH_4$ and F-gases, and ±60 % for $N_2O$ (note that the GCB assesses 1 standard deviation
uncertainty for $CO_2$-FFI as ±5 % and for $CO_2$-LULUCF as ±2.6 $GtCO_2$; Friedlingstein et al., 2022). The total
uncertainty was summed in quadrature, assuming independence of estimates per species/source. Reflecting these
uncertainties, AR6 WGIII reported emissions to two significant figures only. Uncertainties in GWP100 metrics of
roughly ±10 % were not applied (Minx et al., 2021).

This analysis tracks the same compilation of GHGs as in AR6 WGIII. We follow the same approach for estimating
uncertainties and $CO_2$-equivalent emissions. We also use the same type of data sources but make important changes
to the specific selection of data sources to further improve the quality of the data, as suggested in the knowledge gap
discussion of the WGIII report (Dhakal et al., 2022). Instead of using EDGAR data (which are now available as version
8), we use GCB data for $CO_2$-FFI, PRIMAP-hist "CR" data for $CH_4$ and $N_2O$, and atmospheric concentrations with
best-estimate lifetimes for UNFCCC F-gas emissions (Hodnebrog et al., 2020). As in AR6 WGIII we use GCB for
net $CO_2$-LULUCF emissions, taking the average of three bookkeeping models (BLUE by Hansis et al., 2015; H&C
by Houghton and Castanho, 2023; OSCAR by Gasser et al., 2020). Bunker emissions are included but military
emissions excluded (e.g. Bun et al. 2024). For more completeness, this year we also include estimates of $N_2O$ and $CH_4$
emissions from global biomass fires, sourced from GFED.

There are three reasons for these specific data choices. First, national greenhouse gas emissions inventories tend to
use improved, higher-tier methods for estimating emissions fluxes than global inventories such as EDGAR (Dhakal
et al., 2022; Minx et al., 2021). As GCB and PRIMAP-hist "CR" integrate the most recent national inventory
submissions to the UNFCCC, selecting these databases makes best use of country-level improvements in data-
gathering infrastructures. It is important to acknowledge, however, that national inventories differ substantially with
respect to reporting intervals, applied methodologies and emissions factors. Notably, the PRIMAP-hist "CR" dataset
has significantly lower total $CH_4$ emissions relative to both the other datasets reported here, and the global atmospheric
inversion estimates evaluated in this paper. A substantive body of literature has evaluated national level $CH_4$ inversions
versus inventories, finding a tendency for the former to exceed the latter (Deng et al. 2022; Tibrewal et al. 2024;
Janardanan et al. 2024; Scarpelli et al. 2022). Compared to the median of reported inversion models from Deng et al.
2022, PRIMAP-Hist CR reports lower $CH_4$ emissions for India, the EU27+UK, Brazil, Russia and Indonesia, but not
in the case of China and the United States (see Supplement Fig 1).

Second, comprehensive reporting of F-gas emissions has remained challenging in national inventories and may
exclude some military applications (see Minx et al., 2021; Dhakal et al., 2022). However, F-gases are entirely
anthropogenic substances, and their concentrations can be measured effectively and reliably in the atmosphere. We
therefore follow the AR6 WGI approach in making use of direct atmospheric observations.



Third, the choice of GCB data for $CO_2$-FFI means we can integrate its projection of that year's $CO_2$ emissions at the
time of publication (i.e. for 2023). No other dataset except GCB provides projections of $CO_2$ emissions on this time
frame. At this point in the publication cycle (mid-year), the other chosen sources provide data points with a 2-year
time lag (i.e. for 2022). While these data choices inform our overall assessment of GHG emissions, we provide a
comparison across datasets for each emissions category, as well as between our estimates and an estimate derived
from AR6 WGIII-like databases (i.e. EDGAR for $CO_2$-FFI and non-$CO_2$ GHG emissions, GCB for $CO_2$-LULUCF).

**2.2 Updated greenhouse gas emissions**

Updated GHG emission estimates are presented in Fig. 2 and Table 1. Total global GHG emissions were
$55 \pm 5.4$ GtCO$_2$e in 2022, the same as previous high levels in 2019 and 2021. Of this total, $CO_2$-FFI contributed
$37.1 \pm 3$ GtCO2, $CO_2$-LULUCF contributed $4.3 \pm 3$ GtCO2, $CH_4$ contributed $9 \pm 2.7$ GtCO$_2$e, $N_2O$ contributed
$3.1 \pm 1.9$ GtCO$_2$e and F-gas emissions contributed $1.7 \pm 0.51$ GtCO$_2$e. Initial projections indicate that total $CO_2$
emissions remained similar in 2023, with emissions from fossil fuel and industry at $37.5 \pm 3$ and from land-use change
at $4.1 \pm 2.9$ GtCO$_2$ (Friedlingstein et al., 2023; see also Liu et al., 2024; IEA, 2023). Note that ODS F-gases such as
chlorofluorocarbons and hydrochlorofluorocarbons are excluded from national GHG emissions inventories. For
consistency with AR6, they are also excluded here. Including them here would increase total global GHG emissions
by 1.3 GtCO$_2$e in 2022.

Average annual GHG emissions for the decade 2013–2022 were $54 \pm 5.4$ GtCO$_2$e, which is the same as the estimate
from last year for 2012-2021. Average decadal GHG emissions have increased steadily since the 1970s across all
major groups of GHGs, driven primarily by increasing $CO_2$ emissions from fossil fuel and industry but also rising
emissions of $CH_4$ and $N_2O$. Stratospheric ozone-depleting F-gases are regulated under the Montreal Protocol and its
amendments and their emissions have declined substantially since the 1990s, whereas emissions of other F-gases,
regulated under the UNFCCC, have grown more rapidly than other greenhouse gas emissions, but from low levels.
Both the magnitude and trend of $CO_2$ emissions from land-use change remain highly uncertain, with the latest data
indicating an average net flux between 4–5 GtCO$_2$ yr$^{-1}$ for the past few decades.

AR6 WGIII reported total net anthropogenic emissions of $59 \pm 6.6$ GtCO$_2$e in 2019 and decadal average annual
emissions of $56 \pm 6.0$ GtCO$_2$e from 2010–2019. By comparison, our estimates here for the AR6 period sum to
$55 \pm 5.5$ GtCO$_2$e in 2019 and an annual average of $53 \pm 5.5$ GtCO$_2$e for the same decade (2010–2019). The difference
between these figures, including the reduced relative uncertainty range, is partly driven by the substantial revision in
GCB $CO_2$-LULUCF estimates between the 2020 version (used in AR6 WGIII) of 6.6 GtCO$_2$ and the 2022 version
(used here) of 4.6 GtCO$_2$. The main reason for this downward revision comes from updated estimates of agricultural
areas by the FAO, which uses multi-annual land-cover maps from satellite remote sensing, leading to lower emissions
from cropland expansion, particularly in the tropical regions. It is important to note that this change is not a reflection
of changed and improved methodology per se but an update of the resulting estimation due to updates in the available
input data. Second, there are relatively small changes resulting from improvements in datasets since AR6, including



the new addition of global biomass burning (landscape fire) emissions. Datasets impacts are largest for $CH_4$, where
the emission estimate has reduced by 1.6 GtCO$_2$e in 2019. This is related to the switch from EDGAR in AR6 to
PRIMAP-hist CR in this study. EDGAR estimates considerably higher $CH_4$ emissions – from fugitive fossil sources,
as well as the livestock, rice cultivation and waste sectors – compared to country-reported data using higher tier
methods, as compiled in PRIMAP-hist CR (see Sect 2.1). Differences in the remaining gases for 2019 are relatively
small in magnitude (increases in $N_2O$ (+0.42 GtCO$_2$e) and UNFCCC-F-gases (+0.2 GtCO$_2$e) and decreases in CO$_2$-
FFI (−0.8 GtCO$_2$e)). Overall, excluding the change due to $CO_2$-LULUCF and $CH_4$, they impact the total GHG
emissions estimate by −0.21 GtCO$_2$e (roughly 3% of the uncertainty in total greenhouse gas emissions).
The fossil fuel share of global greenhouse gas emissions was approximately 70% in 2022 (GWP100 weighted), based
on the EDGAR v8 dataset (Crippa et al. 2023) and net land use $CO_2$ emissions from the Global Carbon Budget
(Friedlingstein et al. 2023). Non fossil fuel emissions are mostly from land-use change, agriculture, cement production,
waste and F-gas emissions.
New literature not available at the time of the AR6 suggests that increases in atmospheric $CH_4$ concentrations are also
driven by methane emissions from wetland changes resulting from climate change (e.g. Basu et al., 2022; Peng et al.,
2022; Nisbet et al., 2023; Zhang et al., 2023). There is also a possible effect from $CO_3$ fertilisation (Feron et al., 2024;
Hu et al., 2023). Such carbon cycle feedbacks are not considered here as they are not a direct emission from human
activity, yet they will contribute to greenhouse gas concentration rise, forcing and energy budget changes discussed
in the next sections. They will become more important to properly account for in future years.

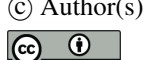

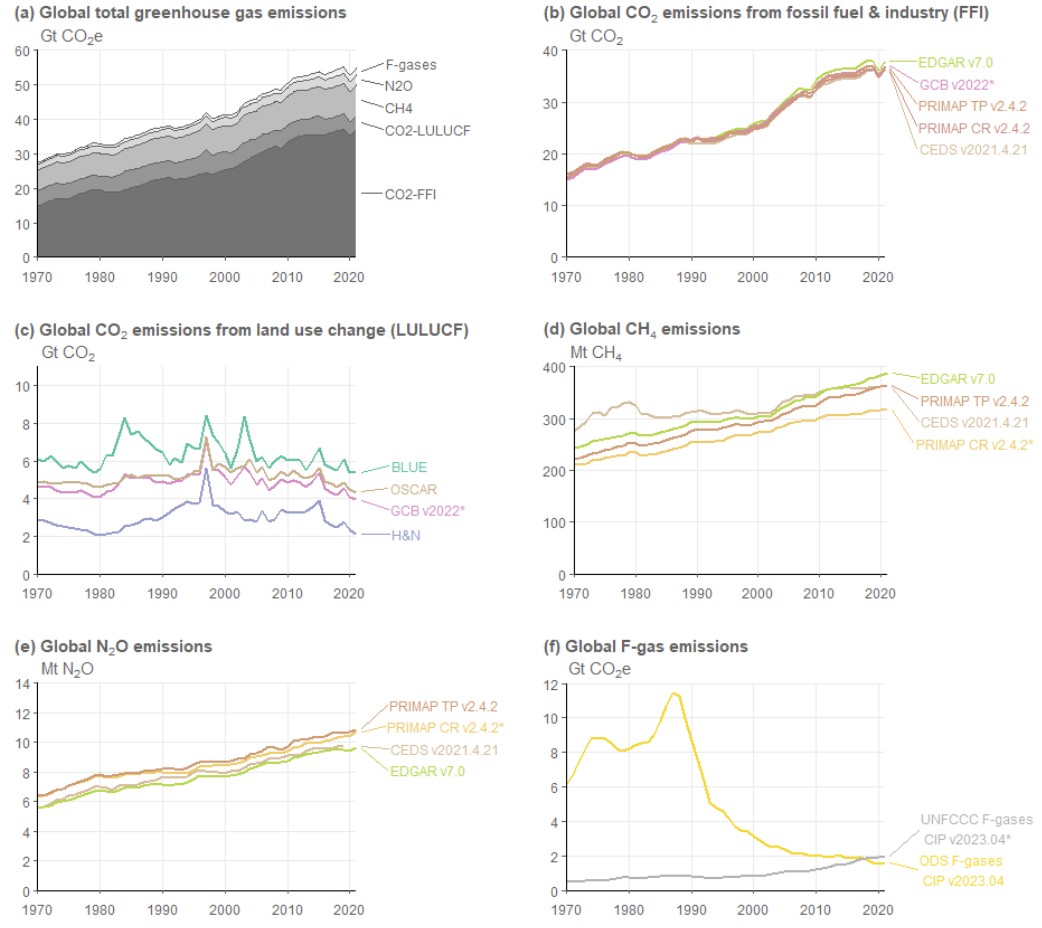

**Figure 2 Annual global anthropogenic greenhouse gas emissions by source, 1970–2022. Refer to Sect. 2.1 for a list of datasets. Datasets with an asterisk (*) indicate the sources used to compile global total greenhouse gas emissions in (a). CO2-equivalent emissions in (a) and (f) are calculated using GWP100 from the AR6 WGI Chap. 7 (Forster et al., 2021). F-gas emissions in (a) comprise only UNFCCC F-gas emissions (see Sect. 2.1 for a list of species). GFED refers to CH4 and N2O emissions from global biomass fires only.**

**Table 1 Global anthropogenic greenhouse gas emissions by source and decade. All numbers refer to decadal averages, except for annual estimates in 2022 and 2023. CO2-equivalent emissions are calculated using GWP100 from AR6 WGI Chap. 7 (Forster et al., 2021). Projections of non-CO2 GHG emissions in 2023 remain unavailable at the time of publication. Uncertainties are ±8 % for CO2-FFI, ±70 % for CO2-LULUCF, ±30 % for CH4 and F-gases, and ±60 % for N2O, corresponding to a 90 % confidence interval. ODS F-gases are excluded, as noted in Sect. 2.1.**

| Units: GtCO2e | 1970-1979 | 1980-1989 | 1990-1999 | 2000-2009 | 2010-2019 | 2013-2022 | 2022 | 2023 (projection) |
|---|---|---|---|---|---|---|---|---|





| GHG | 31±4.2 | 35±4.7 | 40±5.2 | 46±5.2 | 53±5.5 | 54±5.4 | 55±5.4 | |
|---|---|---|---|---|---|---|---|---|
| $CO_2$-FFI | 17.3±1.4 | 20.3±1.6 | 23.6±1.9 | 28.9±2.3 | 35.4±2.8 | 36±2.9 | 37.1±3 | 37.5±3 |
| $CO_2$-LULUCF | 4.6±3.3 | 5.2±3.7 | 5.8±4 | 5.2±3.6 | 5.2±3.5 | 4.7±3.3 | 4.3±3 | 4.1±2.9 |
| $CH_4$ | 6.3±1.9 | 6.9±2.1 | 7.5±2.3 | 8.1±2.4 | 8.8±2.6 | 8.8±2.7 | 9±2.7 | |
| $N_2O$ | 2.1±1.2 | 2.3±1.4 | 2.5±1.5 | 2.7±1.6 | 2.9±1.8 | 3±1.8 | 3.1±1.9 | |
| UNFCCC F-gases | 0.57±0.17 | 0.73±0.22 | 0.67±0.2 | 0.9±0.27 | 1.3±0.39 | 1.5±0.44 | 1.7±0.51 | |


## 2.3 Non-methane short-lived climate forcers

In addition to GHG emissions, we provide an update of anthropogenic emissions of non-methane short-lived climate
forcers (SLCFs) ($SO_2$, black carbon (BC), organic carbon (OC), NOx, volatile organic compounds (VOCs), CO and
$NH_3$). Data is presented in Table 2. HFCs are considered in Sect. 2.2.

Sectoral emissions of SLCFs are derived from two sources. For fossil fuel, industrial, waste and agricultural sectors,
we use the CEDS dataset. CEDS provides global emissions totals from 1750 to 2022 in its most recent version
(v_2024_04_01) (Hoesly et al., 2018; Hoesly & Smith, 2024). No CEDS emissions data are currently available for
2023. The estimate for 2023 was derived by assuming a scaled return to an underlying SSP2-4.5 emissions scenario,
used for inputs to COVID-MIP (Forster et al., 2020, Lamboll et al. 2021). We find that the 2020-2022 emissions trends
comparing CEDS and the COVID-MIP extrapolation are not substantially different (Supplement Fig. S2), so the
COVID-MIP extension to 2023 is justifiable. In Forster et al. (2023), the CEDS dataset was only available to 2019,
so the COVID-MIP extension was used to 2022. Therefore, emissions from 2020 have been revised in this year's
paper with 2020-2022 data now arising from CEDS.

Overall, the net $SO_2$ emissions were similar (within 2 $TgSO_2$, see Supplement Sect. S2) over the 2020-22 period in
the CEDS dataset than our estimate in Forster et al. (2023). The CEDS dataset accounts for the introduction of strict
fuel sulphur controls brought in by the International Maritime Organization on 1 January 2020. Total $SO_2$ emissions
in 2019 were 84.2 $TgSO_2$ (Table 2). The $SO_2$ emissions from international shipping declined by 7.4 $TgSO_2$ from 10.4
$TgSO_2$ in 2019 to 3.0 $TgSO_2$ in 2020, which is close to the expected 8.5 $TgSO_2$ reduction estimated by the IMO,
approximately -80% from the 2019 number, accounting for a 3-month phase in period and COVID-19 changes. Non-
shipping $SO_2$ emissions were impacted slightly by COVID-19, but had rebounded to close to 2019 levels by 2022 in
CEDS.

For biomass burning SLCF emissions, we follow AR6 WGIII (Dhakal et al., 2022) and use GFED (van der Werf et
al., 2017) version 4 with small fires (GFED4s) for 1997 to 2023, with the dataset extended back to 1750 for CMIP6





(van Marle et al., 2017). Estimates from 2017 to 2023 are provisional.  As demonstrated with the update to CEDS
emissions, the potential for both sources of emissions data to be updated in future versions exists, for example with a
planned introduction of GFED5 in preparation for CMIP7.

Using our combined estimate of GFED and CEDS with a 2023 extrapolation, emissions of all SLCFs were reduced in
2022 relative to 2019, but rebounded again in 2023 (Table 2). The primary driver of the increase in 2023 is an
anomalous biomass burning year, mostly related to the unprecedented 2023 Canadian fire season, with a smaller
contribution from a continued recovery from COVID-19. Under these assumptions, 2023 was a record year for
emissions of organic carbon (driven again by a very active biomass burning season) and ammonia (driven by a steady
background increase in agricultural sources, plus a contribution from biomass burning). Causes of the enhanced
burning are not distinguished in the GFED data. Whether human-caused burning, a feedback due to the extreme heat
or naturally occurring, we choose to include them in our tracking, as historical biomass burning emissions inventories
have previously been consistently treated as a forcing (for example in CMIP6), though this assumption may need to
be revisited in the future. . This differs from the treatment of accounting for $CO_2$ and $CH_4$ emissions at present (Sect.
2.2), where we do not include natural emissions in the inventories.  As described in Sect. 4, the treatment of all biomass
burning emissions as a forcing has implications for several categories of anthropogenic radiative forcing. Trends in
SLCFs emissions are spatially heterogeneous (Szopa et al., 2021), with strong shifts in the locations of reductions and
increases over the 2010–2019 decade (Hodnebrog et al. 2024).
**Table 2 Emissions of the major SLCFs in 1750, 2019, 2022 and 2023 from a combination of CEDS and GFED. Emissions of**
**$SO_2$+$SO_4$ use $SO_2$ molecular weights. Emissions of $NO_x$ use $NO_2$ molecular weights. VOCs are for the total mass.**

| Compound | 1750 emissions (Tg yr$^{-1}$) | 2019 emissions (Tg yr$^{-1}$) | 2022 emissions (Tg yr$^{-1}$) | 2023 emissions (Tg yr$^{-1}$) |
|---|---|---|---|---|
| Sulfur dioxide ($SO_2$) + sulfate ($SO_4^{2-}$) | 0.8 | 84.2 | 75.3 | 79.1 |
| Black carbon (BC) | 2.1 | 7.5 | 6.8 | 7.3 |
| Organic carbon (OC) | 15.5 | 34.2 | 25.8 | 40.7 |
| Ammonia ($NH_3$) | 6.6 | 67.6 | 67.3 | 71.1 |
| Oxides of nitrogen ($NO_x$) | 19.4 | 141.7 | 130.4 | 139.4 |
| Volatile organic compounds (VOCs) | 60.9 | 217.3 | 183.9 | 228.1 |
| Carbon monoxide (CO) | 348.4 | 853.8 | 686.4 | 917.5 |


Uncertainties associated with these emission estimates are difficult to quantify. From the non-biomass-burning sectors
they are estimated to be smallest for $SO_2$ (±14 %), largest for black carbon (BC) (a factor of 2) and intermediate for



other species (Smith et al., 2011; Bond et al., 2013; Hoesly et al., 2018). Uncertainties are also likely to increase both
backwards in time (Hoesly et al., 2018) and again in the most recent years. The estimates of non-biomass-burning
emissions for 2023, especially $SO_2$ are highly uncertain, owing to the use of proxy activity data used with a SSP2-4.5
scenario extension (see above). Future updates of CEDS are expected to include uncertainties (Hoesly et al., 2018).
Even though trends over recent years are uncertain, the general decline in some SLCF emissions derived from
inventories punctuated by temporary anomalous years with high biomass burning emissions including 2023 is
supported by MODIS Terra and Aqua aerosol optical depth measurements (e.g. Quaas et al., 2022, Hodnebrog et al

340    2024).

**3 Well-mixed greenhouse gas concentrations**
As in Forster et al. (2023), we report best-estimate global mean concentrations for 52 well-mixed greenhouse gases.
These concentrations are updated to 2023.

As in AR6 and Forster et al. (2023), $CO_2$ mixing ratios were taken from the NOAA Global Monitoring Laboratory
(GML) and are updated here through 2023 (Lan et al., 2024a). As in Forster et al. (2023), $CO_2$ is reported on the
WMO-$CO_2$-X2019 scale, which differs from the WMO-$CO_2$-X2007 scale used in AR6. Prior to the use of NOAA
GML data from 1980 onwards, a conversion is applied to the AR6 $CO_2$ time series to take into account the scale
change using X2019 = 1.00079 * X2007 - 0.142 ppm. Other LLGHG records were compiled from NOAA and AGAGE
global networks or extrapolated from literature. An average of NOAA and AGAGE data were used for $N_2O$, $CH_4$,
CFC-11, CFC-12, CFC-113, $CCl_4$, HCFC-22, HFC-134a, and HFC-125 (Lan et al., 2024b; Dutton et al., 2024; Prinn
et al., 2018), which, along with $CO_2$, account for over 98% of the ERF from well-mixed greenhouse gases. In cases
where no updated information is available, global estimates were extrapolated from Vimont et al. (2022), Western et
al. (2023), or other literature and scaled to be consistent with those reported in AR6. Some extrapolations are based
on data from the mid-2010s (Droste et al., 2020; Laube et al., 2014; Simmonds et al., 2017; Vollmer et al., 2018), but
have an imperceptible effect on the total ERF assessed in Sect. 4, and are included to maintain consistency with AR6.
Mixing ratio uncertainties for 2023 are assumed to be similar to 2019, and we adopt the same uncertainties as assessed
in AR6 WGI.

The global surface mean concentrations of $CO_2$, $CH_4$ and $N_2O$ in 2023 were 419.3 [±0.4] parts per million (ppm),
1922.5 [±3.3] parts per billion (ppb) and 336.9 [±0.4] ppb, respectively. Concentrations of all three major GHGs have
increased since 2019, with $CO_2$ increasing by 9.2 ppm, $CH_4$ by 56 ppb, and $N_2O$ by 4.8 ppb. Increases since 2019 are
consistent with those from the CSIRO network (Francey at al., 1999), which are 9.3 ppm, 55 ppb, and 5.0 ppb for
$CO_2$, $CH_4$, and $N_2O$, respectively. With few exceptions, concentrations of ozone-depleting substances, such as CFC-
11 and CFC-12, continue to decline, while those of replacement compounds (HFCs) have increased. HFC-134a, for
example, has increased 20% since 2019 to 129.5 parts per trillion (ppt). Aggregated across all gases, PFCs have
increased from 109.7 to an estimated 115 ppt $CF_4$-eq from 2019 to 2023, HFCs from 237 to 301 ppt HFC-134a-eq,



while Montreal gases have declined from 1032 to 1004 ppt CFC-12-eq. Mixing ratio equivalents are determined by
the radiative efficiencies of each greenhouse gas from Hodnebrog et al. (2020).

Ozone is an important greenhouse gas with strong regional variation both in the stratosphere and troposphere (Szopa
et al., 2021). Its ERF arising from its regional distribution is assessed in Sect. 4 but following AR6 convention is not
included with the GHGs discussed here. Other non-methane SLCFs are heterogeneously distributed in the atmosphere
and are also not typically reported in terms of a globally averaged concentration. Globally averaged concentrations
for these are normally model-derived, supplemented by local monitoring networks and satellite data (Szopa et al.,

376    2021).


In this update we employ AR6-derived uncertainty estimates and do not perform a new assessment. Table S1 in
Supplement Sect. S3 shows specific updated concentrations for all the GHGs considered.
**4 Effective radiative forcing (ERF)**
ERFs were principally assessed in Chap. 7 of AR6 WGI (Forster et al., 2021), which focussed on assessing ERF from
changes in atmospheric concentrations; it also supported estimates of ERF in Chap. 6 that attributed forcing to specific
precursor emissions (Szopa et al., 2021) and also generated the time history of ERF shown in AR6 WGI Fig. 2.10 and
discussed in Chap. 2 (Gulev et al., 2021). Only the concentration-based estimates are updated herein.

The ERF calculation follows the methodology used in AR6 WGI (Smith et al., 2021) as updated by Forster et al.
(2023). For each category of forcing, a 100 000-member probabilistic Monte Carlo ensemble is sampled to span the
assessed uncertainty range in each forcing. All uncertainties are reported as 5 %–95 % ranges and provided in square
brackets. The methods are all detailed in the Supplement, Sect. S4.

The summary results for the anthropogenic constituents of ERF and solar irradiance in 2023 relative to 1750 are shown
in Fig. 3a. In Table 3 these are summarised alongside the equivalent ERFs from AR6 (1750–2019) and last year's
Climate Indicators update (1750-2022). Fig. 3b shows the time evolution of ERF from 1750 to 2023.

**Table 3 Contributions to anthropogenic effective radiative forcing (ERF) for 1750–2023 assessed in this section. Data is for**
**single year estimates unless specified. All values are in watts per square metre (W m$^{-2}$), and 5 %–95 % ranges are in square**
**brackets. As a comparison, the equivalent assessments from AR6 (1750–2019) and last year's Climate Indicators (1750-**
**2022) are shown. Solar ERF is included and unchanged from AR6, based on the most recent solar cycle (2009–2019), thus**
**differing from the single-year estimate in Fig. 3a. Volcanic ERF is excluded due to the sporadic nature of eruptions.**

| Forcer | 1750-2019 [W m$^{-2}$] (AR6) | 1750-2022 [W m$^{-2}$] (Forster et al., 2023) | 1750-2023 [W m$^{-2}$] | Reason for change since last year |
|--------|------------------------------|-----------------------------------------------|------------------------|-----------------------------------|
| $CO_2$ | 2.16 [1.90 to 2.41] | 2.25 [1.98 to 2.52] | 2.28 [2.01 to 2.56] | Increases in GHG |





| | | | | concentrations resulting from increases in emissions |
|---|---|---|---|---|
| CH$_4$ | 0.54 [0.43 to 0.65] | 0.56 [0.45 to 0.67] | 0.56 [0.45 to 0.68] | |
| N$_2$O | 0.21 [0.18 to 0.24] | 0.22 [0.19 to 0.25] | 0.22 [0.19 to 0.26] | |
| Halogenated GHGs | 0.41 [0.33 to 0.49] | 0.41 [0.33 to 0.49] | 0.41 [0.33 to 0.49] | |
| Ozone | 0.47 [0.24 to 0.71] | 0.48 [0.24 to 0.72] | 0.51 [0.25 to 0.76] | Increase in precursors (CO, VOC, CH$_4$) |
| Stratospheric water vapour | 0.05 [0.00 to 0.10] | 0.05 [0.00 to 0.10] | 0.05 [0.00 to 0.10] | |
| Aerosol-radiation interactions | -0.22 [-0.47 to +0.04] | -0.21 [-0.42 to 0.00] | -0.26 [-0.50 to -0.03] | Large increases in biomass burning aerosol in 2023; continued recovery from COVID-19; drop in sulphur from shipping |
| Aerosol-cloud interactions | -0.84 [-1.45 to -0.25] | -0.77 [-1.33 to -0.13] | -0.91 [-1.80 to -0.27] | |
| Land use (surface albedo changes and effects of irrigation) | -0.20 [-0.30 to -0.10] | -0.20 [-0.30 to -0.10] | -0.20 [-0.31 to -0.10] | |
| Light-absorbing particles on snow and ice | 0.08 [0.00 to 0.18] | 0.06 [0.00 to 0.14] | 0.08 [0.00 to 0.17] | Rebound in BC emissions from biomass burning |
| Contrails and contrail-induced cirrus | 0.06 [0.02 to 0.10] | 0.05 [0.02 to 0.09] | 0.05 [0.02 to 0.09] | Estimates of aviation activity are rebounding since the pandemic but still below 2019 levels in 2023 |
| Total anthropogenic | 2.72 [1.96 to 3.48] | 2.91 [2.19 to 3.63] | 2.79 [1.78 to 3.60] | Possible strong aerosol forcing in 2023 partly offset by increases in GHG and ozone forcing |
| Solar irradiance | 0.01 [-0.06 to 0.08] | 0.01 [-0.06 to 0.08] | 0.01 [-0.06 to 0.08] | |





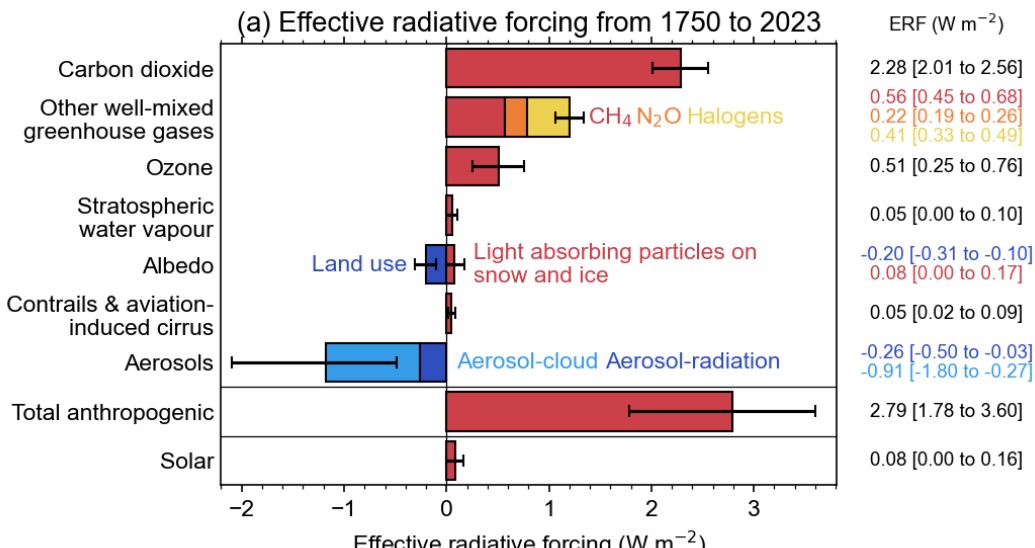


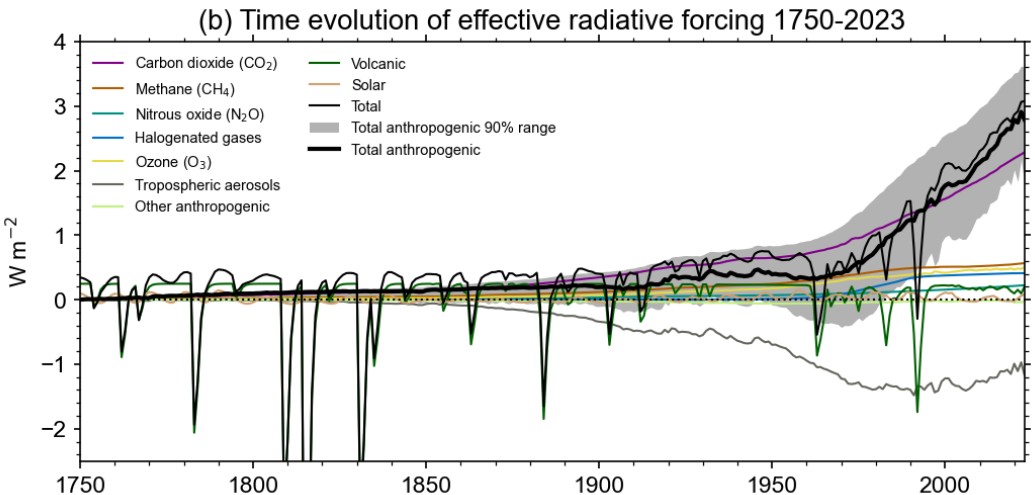


**Figure 3 Effective radiative forcing from 1750–2023. (a) 1750–2023 change in ERF, showing best estimates (bars) and 5 %–95 % uncertainty ranges (lines) from major anthropogenic components to ERF, total anthropogenic ERF and solar forcing. Note that solar forcing in 2023 is a single-year estimate. (b) Time evolution of ERF from 1750 to 2023. Best estimates from major anthropogenic categories are shown along with solar and volcanic forcing (thin coloured lines), total (thin black line), and anthropogenic total (thick black line). The 5 %–95 % uncertainty in the anthropogenic forcing is shown by grey shading.**

Total anthropogenic ERF has increased to 2.79 [1.78 to 3.60] W m$^{-2}$ in 2023 relative to 1750, compared to 2.72 [1.96 to 3.48] W m$^{-2}$ for 2019 relative to 1750 in AR6. The estimate of ERF for 2023 is lower than the 2.91 [2.19 to 3.63]



412 W m$^{-2}$ in 2022 evaluated in last year's Indicators. The main reason for the decline in 2023 relative to 2022 is a very

413 strong contribution from biomass burning aerosol in 2023, particularly organic carbon emissions which strengthened

414 the negative aerosol ERF (see also Sect. 2.3). Sulphur emissions from shipping have declined since 2020, weakening

415 the aerosol ERF and adding around +0.1 W m$^{-2}$ over 2020 to 2023 (Gettelman et al., 2024; see Supplement Sect.

416 S4.2.2). However, the strengthened negative ERF from increased biomass burning likely dominated the effect of

417 reduced shipping emissions. As discussed in Sect. 2, it is not easy to determine how much of the biomass burning

418 contribution is from natural wildfires in response to 2023's anomalously warm year, which would be a climate

419 feedback rather than a forcing. We follow the convention of CMIP and count all biomass burning emissions as

420 anthropogenic, though this assumption may need revision in future. The approach of including all biomass burning

421 aerosols is consistent with reporting ERF based on concentration increase of GHGs independent of whether $CO_2$ and

422 $CH_4$ are caused by anthropogenic emissions or a smaller part is caused by any feedbacks such as from biomass burning

423 fires or wetlands. However, changes in mineral dust and sea salt are not included in ERF of aerosols and any changes

424 are interpreted as yearly variations or related to feedbacks.

425

426 The relative uncertainty in the total ERF was at the lowest reported in 2022, see Table 3, but with the strengthening

427 of the aerosol ERF due to biomass additional burning, the relative uncertainty in total ERF for 2023 is higher than in

428 2019 reported in AR6 (Forster et al., 2021). Despite the strong aerosol forcing in 2023, decadal trends in anthropogenic

429 ERF remain high, and are over 0.6 W m$^{-2}$ per decade. These are discussed further in Sect. 7.3.

430

431 The ERF from well-mixed GHGs is 3.45 [3.14 to 3.75] W m$^{-2}$ for 1750–2022, of which 2.25 W m$^{-2}$ is from $CO_2$,

432 0.56 W m$^{-2}$ from $CH_4$, 0.22 W m$^{-2}$ from $N_2O$ and 0.41 W m$^{-2}$ from halogenated gases. This is an increase from 3.32

433 [3.03 to 3.61] W m$^{-2}$ for 1750–2019 in AR6. ERFs from $CO_2$, $CH_4$ and $N_2O$ have all increased since the AR6 WG1

434 assessment for 1750–2019, owing to increases in atmospheric concentrations.

435

436 The total aerosol ERF (sum of the ERF from aerosol–radiation interactions (ERFari) and aerosol–cloud interactions

437 (ERFaci)) for 1750–2023 is −1.18 [−2.10 to −0.49] W m$^{-2}$ compared to −0.98 [−1.58 to −0.40] W m$^{-2}$ in Forster et

438 al. (2023) and −1.06 [−1.71 to −0.41] W m$^{-2}$ assessed for 1750–2019 in AR6 WG1. This counters a recent trend of

439 reductions in aerosol forcing, and is related in most part to 2023 being an extremely active biomass burning season.

440 Most of this reduction is from ERFaci, which is determined to be −0.91 [−1.80 to −0.27] W m$^{-2}$ in 2023 compared to

441 −0.77 [−1.33 to −0.23] W m$^{-2}$ for 1750–2022 (Forster et al. 2023) and −0.84 [−1.45 to −0.25] W m$^{-2}$ in AR6 for 1750–

442 2019. ERFari for 1750–2023 is −0.26 [−0.50 to −0.03] W m$^{-2}$, stronger than the −0.21 [−0.42 to 0.00] W m$^{-2}$ for 1750–

443 2022 and the −0.22 [−0.47 to 0.04] W m$^{-2}$ assessed for 1750–2019 in AR6 WG1 (Forster et al., 2021). The largest

444 contributions to ERFari are from $SO_2$ (primary source of sulfate aerosol; −0.24 W m$^{-2}$), BC (+0.16 W m$^{-2}$), OC

445 (−0.11 W m$^{-2}$) and $NH_3$ (primary source of nitrate aerosol; −0.04 W m$^{-2}$). ERFari also includes terms from $CH_4$, $N_2O$,

446 VOCs and NOx which are small.


448 Ozone ERF is determined to be 0.51 [0.25 to 0.76] W m$^{-2}$ for 1750–2023, slightly higher than the  the AR6 assessment

449 of 0.47 [0.24 to 0.71] W m$^{-2}$ for 1750–2019. This is due to the increase in emissions of some of its precursors (CO,



VOC, $CH_4$), but this result is highly uncertain since the preliminary OMI/MLS satellite data indicate tropospheric ozone
burden is stable from 2020 to 2023 (meaning that the 2023 level does not reach the 2019 one) which could be partly
due to the 2020-2023 levels of tropospheric $NO_2$ than the pre-COVID levels (OMI data from Krotkov et al. 2019).
Land-use forcing and stratospheric water vapour from methane oxidation are unchanged (to two decimal places) since
AR6. BC emissions have increased between 2022 and 2023, and were similar to 2019 levels in 2023 resulting inERF
from light-absorbing particles on snow and ice being 0.08 [0.00 to 0.17] $W\,m^{-2}$ for 1750–2023, similar to AR6.  We
determine from provisional data that aviation activity in 2023 had not yet returned to pre-COVID levels. Therefore,
ERF from contrails and contrail-induced cirrus remains lower than AR6, at 0.05 [0.02 to 0.09] $W\,m^{-2}$ in 2023
compared to 0.06 [0.02 to 0.10] $W\,m^{-2}$ in 2019.

The headline assessment of solar ERF is unchanged, at 0.01 [−0.06 to +0.08] $W\,m^{-2}$ from pre-industrial to the 2009–
2019 solar cycle mean. Separate to the assessment of solar forcing over complete solar cycles, we provide a single-
year solar ERF for 2023 of 0.08 [0.00 to +0.16] $W\,m^{-2}$. This is higher than the single-year estimate of solar ERF for
2019 (a solar minimum) of −0.02 [−0.08 to 0.06] $W\,m^{-2}$.

Volcanic ERF is included in the overall time series (Fig. 3b) but following IPCC convention we do not provide a
single-year estimate for 2023 given the sporadic nature of volcanoes. Alongside the time series of stratospheric aerosol
optical depth derived from proxies and satellite products, for 2022 and 2023 we include the stratospheric water vapour
contribution from the Hunga Tonga-Hunga Ha'apai (HTHH) eruption derived from Microwave Limb Sounder (MLS)
data.

Stratospheric water vapour forcing is estimated to be +0.14 $W\,m^{-2}$ in 2022 and +0.18 $W\,m^{-2}$ in 2023, and in 2023
almost totally offsets the negative forcing from stratospheric aerosol.

**5 Earth energy imbalance**
The Earth energy imbalance (EEI), assessed in Chap. 7 of AR6 WGI (Forster et al., 2021), provides a measure of
accumulated surplus energy (heating) in the climate system, and is hence an essential indicator to monitor the current
and future status of global warming. It represents the difference between the radiative forcing acting to warm the
climate and Earth's radiative response, which acts to oppose this warming. On annual and longer timescales, the global
Earth heat inventory changes associated with EEI are dominated by the changes in global ocean heat content (OHC),
which accounts for about 90 % of global heating since the 1970s (Forster et al., 2021). This planetary heating results
in changes in all components of the Earth system such as sea level rise, ocean warming, ice loss, rise in temperature
and water vapor in the atmosphere, changes in ocean and atmospheric circulation, ice loss and permafrost thawing
(e.g. Cheng et al., 2022; von Schuckmann et al., 2023a), with adverse impacts for ecosystems and human systems
(Douville et al., 2021; IPCC, 2022).





On decadal timescales, changes in global surface temperatures (Sect. 5) can become decoupled from EEI by ocean
heat rearrangement processes (e.g. Palmer and McNeall, 2014; Allison et al., 2020). Therefore, the increase in the
Earth heat inventory provides a robust indicator of the rate of global change on interannual-to-decadal timescales
(Cheng et al., 2019; Forster et al., 2021; von Schuckmann et al., 2023a). AR6 WGI found increased confidence in the
assessment of change in the Earth heat inventory compared to previous IPCC reports due to observational advances
and closure of the energy and global sea level budgets (Forster et al., 2021; Fox-Kemper et al., 2021).

AR6 estimated that EEI increased from 0.50 [0.32–0.69] W m$^{-2}$ during the period 1971–2006 to 0.79 [0.52–
1.06] W m$^{-2}$ during the period 2006–2018 (Forster et al., 2021). The contributions to increases in the Earth heat
inventory throughout 1971–2018 remained stable: 91 % for the full-depth ocean, 5 % for the land, 3 % for the
cryosphere and about 1 % for the atmosphere (Forster et al., 2021). Two recent studies demonstrated independently
and consistently that since 1960, the warming of the world ocean has accelerated at a relatively consistent pace
of 0.15 ± 0.05 W m$^{-2}$ per decade (Minière et al., 2023; Storto and Yang, 2024), while the land, cryosphere, and
atmosphere have exhibited an accelerated pace of 0.013 ± 0.003 W m$^{-2}$ per decade (Minière et al., 2023). The
increase in EEI over the most recent quarter of a decade (Fig. 4) has also been reported by Cheng et al. (2019), von
Schuckmann et al. (2020, 2023a), Loeb et al. (2021), Hakuba et al. (2021), Kramer et al. (2021) Raghuraman et al.
(2021) and Minère et al. (2023). Drivers for the observed increase over the most recent period (i.e. past 2 decades) are
discussed to be linked to rising concentrations of well-mixed greenhouse gases and recent reductions in aerosol
emissions (Raghuraman et al., 2021; Kramer et al., 2021; Hansen et al., 2023 ), and to an increase in absorbed solar
radiation associated with decreased reflection by clouds and sea-ice and a decrease in outgoing longwave radiation
(OLR) due to increases in trace gases and water vapor (Loeb et al., 2021) . The degree of contribution from the
different drivers is uncertain and still under active investigation.


We carry out an update to the AR6 estimate of changes in the Earth heat inventory based on updated observational
time series for the period 1971–2020 (Table 4 and Fig. 4). Time series of heating associated with loss of ice and
warming of the atmosphere and continental land surface are obtained from the recent Global Climate Observing
System (GCOS) initiative (von Schuckmann et al., 2023b; Adusumilli et al., 2022; Cuesta-Valero et al., 2023;
Vanderkelen and Thiery, 2022; Nitzbon et al., 2022; Kirchengast et al., 2022). We use the original AR6 time series
ensemble OHC time series for the period 1971–2018 and then an updated five-member ensemble for the period 2019–
2023. We "splice" the two sets of time series by adding an offset as needed to ensure that the 2018 values are identical.
The AR6 heating rates and uncertainties for the ocean below 2000 m are assumed to be constant throughout the period.
The time evolution of the Earth heat inventory is determined as a simple summation of time series of atmospheric
heating; continental land heating; heating of the cryosphere; and heating of the ocean over three depth layers: 0–700,
700–2000 and below 2000 m (Fig. 4a). While von Schuckmann et al. (2023a) have also quantified heating of
permafrost and inland lakes and reservoirs, these additional terms are very small and are omitted here for consistency
with AR6 (Forster et al., 2021).




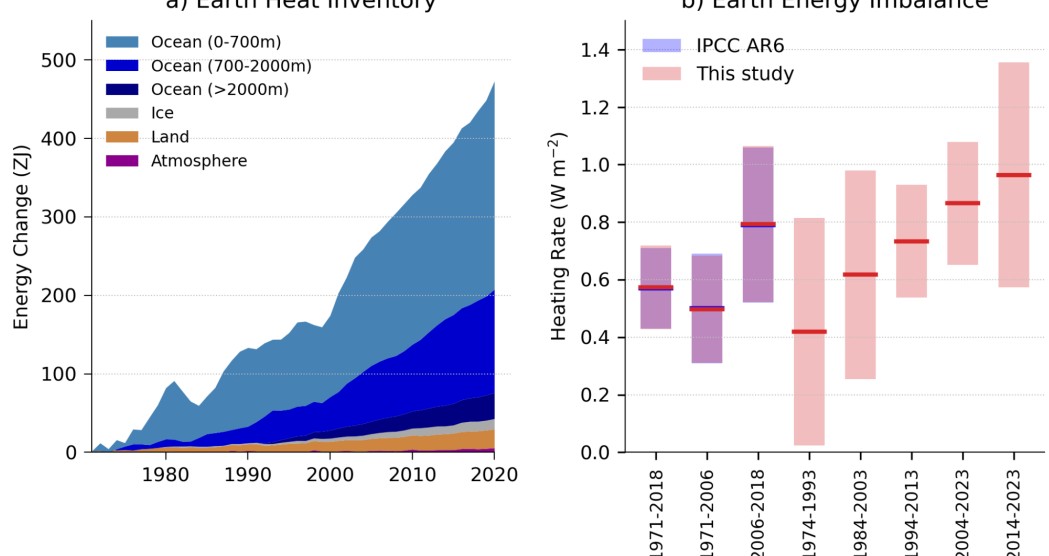


**Figure 4 (a) Observed changes in the Earth heat inventory for the period 1971–2020, with component contributions as**
**indicated in the figure legend. (b) Estimates of the Earth energy imbalance for IPCC AR6 assessment periods, for**
**consecutive 20-year periods and the most recent decade. Shaded regions indicate the *very likely* range (90 % to 100 %**
**probability). Data use and approach are based on the AR6 methods and further described in the Supplement Sect. 5**
**Materials.**
In our updated analysis, we find successive increases in EEI for each 20-year period since 1974, with an estimated
value of 0.42 [0.02 to 0.81] W m$^{-2}$ during 1974–1993 that more than doubled to 0.87 [0.65 to 1.08] W m$^{-2}$ during
2004–2023 (Fig. 4b). In addition, there is some evidence that the warming signal is propagating into the deeper ocean
over time, as seen by a robust increase of deep (700–2000 m) ocean warming since the 1990s (von Schuckmann et al.,
2020; 2023; Cheng et al., 2019, 2022). The model simulations qualitatively agree with the observational evidence (e.g.
Gleckler et al., 2016; Cheng et al., 2019), further suggesting that more than half of the OHC increase since the late
1800s occurs after the 1990s.

The update of the AR6 assessment periods to end in 2023 results in systematic increases of EEI: 0.65 W m$^{-2}$ during
1976–2023 compared  to 0.57 W m$^{-2}$ during 1971–2018; and 0.96 W m$^{-2}$ during 2011–2023 compared to 0.79 W m$^{-2}$
2006–2018 (Table 4). The trend and interannual variability of EEI can largely be explained by a combination of
surface temperature changes and radiative forcing (Hodnebrog et al., 2024), although there was a jump in 2023 which
is still being investigated (Hansen et al., 2023).




**Table 4 Estimates of the Earth energy imbalance (EEI) for AR6 and the present study.**

| Time Period | Earth energy imbalance (W m$^{-2}$). Square brackets [show 90% confidence intervals]. | |
| --- | --- | --- |
| | **IPCC AR6** | **This Study** |
| 1971-2018 | 0.57 [0.43 to 0.72] | 0.57 [0.43 to 0.72] |
| 1971-2006 | 0.50 [0.32 to 0.69] | 0.50 [0.31 to 0.68] |
| 2006-2018 | 0.79 [0.52 to 1.06] | 0.79 [0.52 to 1.07] |
| 1976-2023 | - | 0.65 [0.48 to 0.82] |
| 2011-2023 | - | 0.96 [0.67 to 1.26] |


**6 Global surface temperatures**
AR6 WGI Chap. 2 assessed the 2001–2020 globally averaged surface temperature change above an 1850–1900
baseline to be 0.99 [0.84 to 1.10] °C and 1.09 [0.95 to 1.20] °C for 2011–2020 (Gulev et al., 2021). Updated estimates
to 2022 of 1.15 [1.00–1.25] °C were given in AR6 SYR (Lee et al., 2023), matching the estimate in Forster et al.
550    (2023).


There are choices around the methods used to aggregate surface temperatures into a global average, how to correct for
systematic errors in measurements, methods of infilling missing data, and whether surface measurements or
atmospheric temperatures just above the surface are used. These choices, and others, affect temperature change
estimates and contribute to uncertainty (IPCC AR6 WGI Chap. 2, Cross Chap. Box 2.3, Gulev et al., 2021). The
methods chosen here closely follow AR6 WGI and are presented in the Supplement, Sect. S6. Confidence intervals
are taken from AR6 as only one of the employed datasets regularly updates ensembles (see Supplement, Sect. S6).

Based on the updates available as of March 2024, the change in global surface temperature from 1850–1900 to 2014–
2023 is presented in Fig. 5. These data, using the same underlying datasets and methodology as AR6, give 1.19 [1.06–
1.30] °C, an increase of 0.10 °C within three years from the 2011–2020 value reported in AR6 WGI (Table 5) and
0.09 °C from the 2011–2020 value in the most recent dataset versions. The change from 1850–1900 to 2004–2023
was 1.05 [0.90–1.16] °C, 0.07 °C higher than the value reported in AR6 WGI from three years earlier. These changes,
although amplified somewhat by the exceptionally warm 2023, are broadly consistent with typical warming rates over
the last few decades, which were assessed in AR6 as 0.76 °C over the 1980–2020 period (using ordinary-least-square
linear trends) or 0.019 °C per year (Gulev et al., 2021). They are also broadly consistent with projected warming rates



from 2001–2020 to 2021–2040 reported in AR6, which are in the order of 0.025 °C per year under most scenarios
(Lee et al., 2021). See Sect. 7.4 for further discussion of trends.

**Table 5 Estimates of global surface temperature change from 1850–1900 [*very likely* (90 %–100 % probability) ranges] for**
**IPCC AR6 and the present study.**

| Time period | Temperature change from 1850-1900 (°C) | |
|---|---|---|
| | IPCC AR6 | This study |
| Global, most recent 10 years | 1.09 [0.95 to 1.20] (to 2011-2020) | 1.19 [1.06 to 1.30] (to 2014-2023) |
| Global, most recent 20 years | 0.99 [0.84 to 1.10] (to 2001-2020) | 1.05 [0.90 to 1.16] (to 2004-2023) |
| Land, most recent 10 years | 1.59 [1.34 to 1.83] (to 2011-2020) | 1.71 [1.41 to 1.94] (to 2014-2023) |
| Ocean, most recent 10 years | 0.88 [0.68 to 1.01] (to 2011-2020) | 0.97 [0.77 to 1.09] (to 2014-2023) |


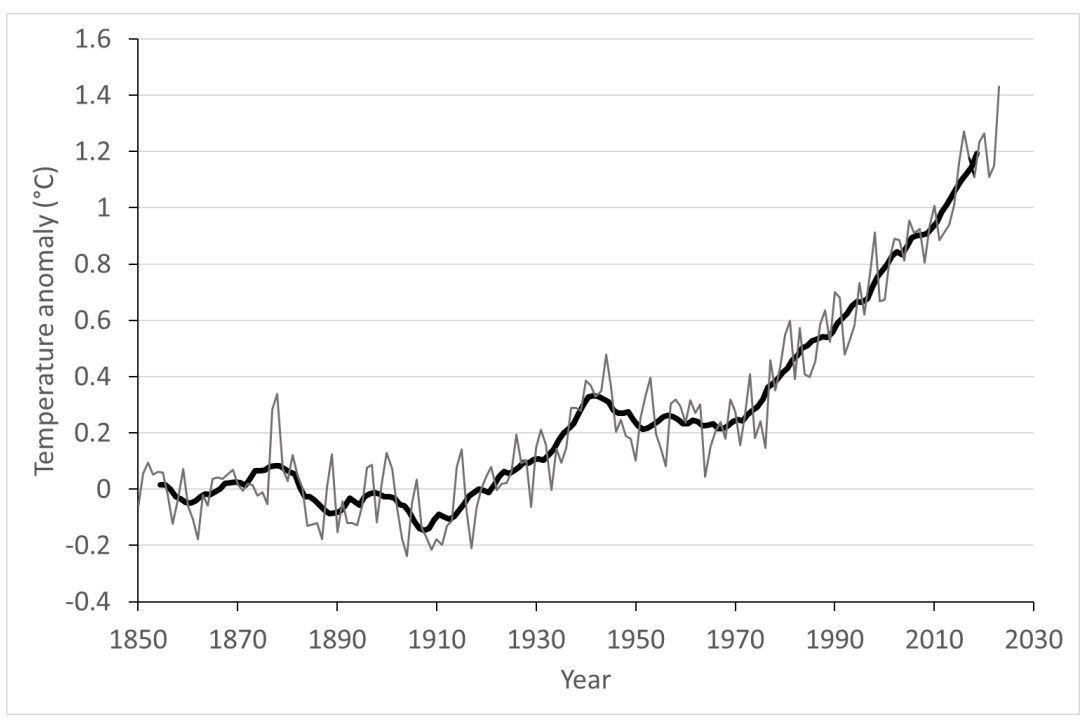







**Figure 5 Annual (thin line) and decadal (thick line) means of global surface temperature (expressed as a change from the**
**1850–1900 reference period).**

The global surface temperature in 2023 was 1.43 [1.32 to 1.53] °C above the 1850-1900 average in the multi-data set
mean used here. This is similar to the combined estimate from six datasets quoted in the 2023 WMO State of the
Climate report 1.45 [1.33 to 1.57] °C (WMO, 2024). As seen in Fig. 5 and discussed in Sect. 7.3, this is considerably
above the human induced warming estimate, indicating a significant role for internal variability.
**7 Human-induced global warming**
Human-induced warming, also known as anthropogenic warming, refers to the component of observed global surface
temperature increase attributable to both the direct and indirect effects of human activities, which are typically grouped
as follows: well-mixed greenhouse gases (consisting of $CO_2$, $CH_4$, $N_2O$ and F-gases) and other human forcings
(consisting of aerosol–radiation interaction, aerosol–cloud interaction, black carbon on snow, contrails, ozone,
stratospheric $H_2O$ and land use) (Eyring et al., 2021). The remaining contributors to total warming are natural
consisting of both natural forcings (such as solar and volcanic activity) and internal variability of the climate system
(such as variability related to El Niño/La Nina events).

While total warming, the observed temperature change resulting from both natural and human influences, is the
quantity more directly related to climate impacts and therefore particularly relevant for adaptation, mitigation efforts
focus on limiting human-induced warming, which better represents the state of long-term climate averages. Further,
as attribution analysis allows human-induced warming to be disentangled from possible contributions from natural
sources, it avoids misperception about short-term fluctuations in temperature, for example in relation to El Niño/La
Nina events.

An assessment of human-induced warming was therefore provided in two reports within the IPCC's 6th assessment
cycle: first in SR1.5 in 2018 [Chap. 1 Sect. 1.2.1.3 and Fig. 1.2 (Allen et al., 2018), summarised in the Summary for
Policymakers (SPM) Sect. A.1 and Fig. SPM.1 (IPCC, 2018)] and second in AR6 in 2021 [WGI Chap. 3 Sect. 3.3.1.1.2
and Fig. 3.8 (Eyring et al., 2021), summarised in the WGI Summary for Policymakers (SPM) Sect. A.1.3 and Fig.
SPM.2 (IPCC, 2021b), and quoted again without any updates in SYR Sect. 2.1.1 and Fig. 2.1 (IPCC,2023a) and SYR
Summary for Policymakers (SPM) Sect. A.1.2. (IPCC 2023b).
**7.1 Warming period definitions in the IPCC Sixth Assessment cycle**
Temperature increases are defined relative to a baseline; IPCC assessments typically use the 1850–1900 average
temperature, as a proxy for the climate in pre-industrial times, referred to as the period before 1750 (see AR6 WGI
Cross Chapter Box 1.2).



Tracking progress towards the long-term global goal to limit warming, in line with the Paris Agreement, requires the
assessment of both what the current level of global surface temperatures are and whether a level of global warming,
such as 1.5°C, is being reached. Definitions for these were not specified in the Paris Agreement, and several ways of
tracking levels of global warming are in use (Betts et al. 2023); here we focus on those adopted within the IPCC's
AR6 (Fig. 6). When determining whether warming thresholds have been passed, both AR6 and SR1.5 adopted
definitions that depend on future warming; in practice, levels of current warming were therefore reported in AR6 and
SR.15 using additional definitions that circumvented the need to wait for observations of the future climate. AR6
defined crossing-time for a level of global warming as the midpoint of the first 20-year period during which the average
*observed* warming for that period, in GSAT, exceeds that level of warming (see AR6 WGI Chapter 2 Box 2.3). It then
reported current levels of both *observed* and *human-induced* warming as their averages over the most recent decade
(see AR6 WGI Chapter 3 Sect. 3.3.1.1.2). This still effectively gives the warming level with a crossing time 5 years
in the past, so would need to be combined with a projection of temperature change over the next decade to give a 20-
year mean with crossing time at the current year (Betts et al., 2023); we do not focus on this here due to the need for
further examination of methods and implications. SR1.5 defined the current level of warming as the average *human-*
*induced* warming, in GMST, of a 30-year period centred on the current year, extrapolating any multidecadal trend into
the future if necessary (see SR1.5 Chapter 1 Sect. 1.2.1). If the multidecadal trend is interpreted as being linear, this
definition of current warming is equivalent to the end-point of the trend line through the most recent 15 years of
human-induced warming, and therefore depends only on historical warming. This interpretation produces results that
are almost all identical to the present-day single-year value of human-induced warming (see Fig. 5, results in Sect.
7.3, and Supplement Sect. S7.3), so in practice the attribution assessment in SR1.5 was based on the single-year
attributed warming calculated using the Global Warming Index, not the trend-based definition.

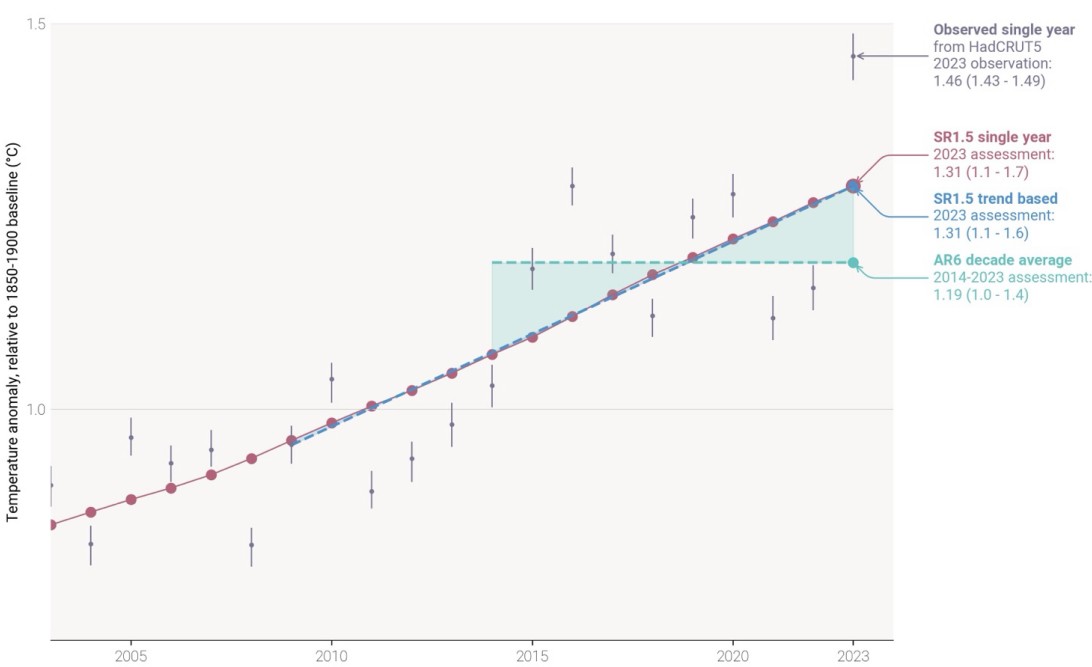

**Figure 6 Anthropogenic warming period definitions adopted in the IPCC Sixth assessment cycle. A single sampled timeseries of anthropogenic warming is shown in red (in this case from the GWI method - see Supplement Sect. S7). Single-year warming is given by the annual values of this timeseries. The AR6 decade average warming is given by the average of the 10 most-recent single year anthropogenic warming values; this is depicted by the green dashed line with shading between this and the red single year values; the decade-average value for 2014-2023 is given by the green dot. SR1.5 trend-based warming is given by the end-point of the linear trend line through the 15 most-recent single year anthropogenic warming values; this is depicted by the blue dashed line with shading between this and the red single-year values; the trend-based value for 2023 is given by the blue dot. Reference observations of GMST are provided from HadCRUT5, with 5-95% uncertainty range. The single-year, trend-based, and decade-average calculations are applied at the level of the individual ensemble members for each attribution method; percentiles of those ensemble results provide central estimates and uncertainty ranges for each method, and the multi-method assessment combines those into the final assessment results with uncertainty (as described in Supplement Sect. S7.4); for reference, the assessment results for 2023 provided in Sect. 7.3 are annotated in the figure (though the data in the figure does not correspond to the final assessment results).**

## 7.2 Updated assessment approach of human-induced warming to date

This paper provides an update of the AR6 WGI and SR1.5 human-induced warming assessments, including for completeness all three definitions (AR6 decade-average, SR1.5 trend-based, and SR1.5 single-year). The 2023 updates in this paper follow the same methods and process as the 2022 updates provided in Forster et al. (2023). Global mean surface temperature is adopted as the definition of global surface temperature (see Supplement Sect. S7.1). The three



attribution methods used in AR6 are retained: the Global Warming Index (GWI) (building on Haustein et al., 2017),
regularised optimal fingerprinting (ROF) (as in Gillett et al., 2021) and kriging for climate change (KCC) (Ribes et
al., 2021). Details of each method, their different uses in SR1.5 and AR6, and any methodological changes, are
provided in Supplement Sect. S7.2; method-specific results are also provided in Supplement Sect. S7.3. The overall
estimate of attributed global warming for each definition (decade-average, trend-based, and single-year), is based on
a multi-method assessment of the three attribution methods (GWI, KCC, ROF); the best estimate is given as the
0.01°C-precision mean of the 50th percentiles from each method, and the *likely* range is given as the smallest 0.1°C-
precision range that envelops the 5th to 95th percentile ranges of each method. This assessment approach is identical
to last year's update (Forster et al. (2023)); it is directly traceable to and fully consistent with the assessment approach
in AR6, though it has been extended in ways that are explained in Supplement Sect. S7.4.

**7.3 Results**
Results are summarised in Table 6 and Figs. 6 and 7. Method-specific contributions to the assessment results, along
with time series, are given in the Supplement, Sect. S7.3. Where results reported in GSAT differ from those reported
in GMST (see Supplement Sect. S7.1), the additional GSAT results are given in Supplement Sect. S7.3.



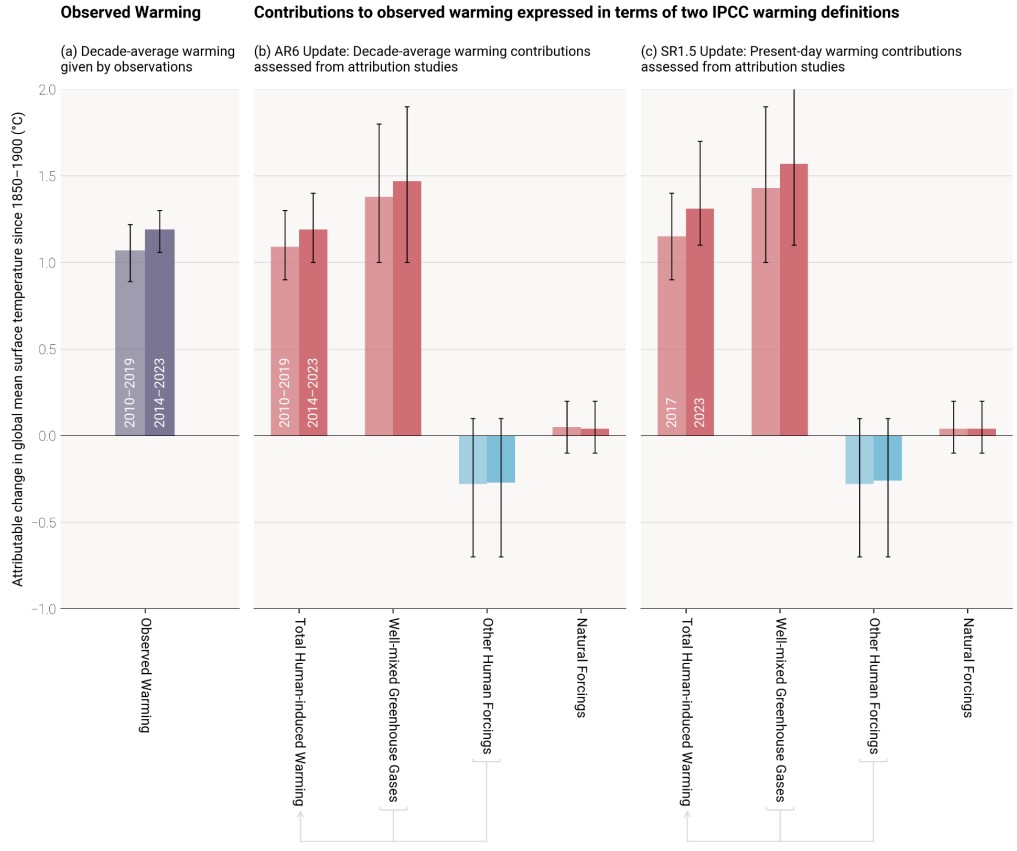






**Figure 7 Updated assessed contributions to observed warming relative to 1850–1900; see AR6 WGI SPM.2. Results for all**
**time periods in this figure are calculated using updated datasets and methods. To show how these updates have affected the**
**previous assessments, the 2010–2019** *decade-average* **assessed results repeat the AR6 2010–2019 assessment, and the 2017**
*single-year* **assessed results repeat the SR1.5 2017 assessment. The 2014–2023** *decade-average* **and 2023** *single-year* **results**
**are this year's updated assessments for AR6 and SR1.5, respectively. For each double bar, the lighter and darker shading**
**refers to the earlier and later period, respectively. Panel (a) shows updated observed global warming from Sect. 6, expressed**
**as total global mean surface temperature (GMST), due to both anthropogenic and natural influences. Whiskers give the**
**"***very likely***" range. Panels (b) and (c) show updated assessed contributions to warming, expressed as global mean surface**
**temperature (GMST), from natural forcings and total human-induced forcings, which in turn consist of contributions from**
**well-mixed greenhouse gases and other human forcings. Whiskers give the "***likely***" range.**
**Table 6 Updates to assessments in the IPCC 6th assessment cycle of warming attributable to multiple influences. Estimates**
**of warming attributable to multiple influences, in °C, relative to the 1850–1900 baseline period. Results are given as best**
**estimates, with the** *likely* **range in brackets, and reported as global mean surface temperature (GMST). Results from the**
**IPCC 6th assessment cycle, for both AR6 and SR1.5, are quoted in columns labelled (i) and are compared with repeat**
**calculations in columns labelled (ii) for the same period using the updated methods and datasets in order to see how**
**methodological and dataset updates alone would change previous assessments. Assessments for the updated periods are**
**reported in columns labelled (iii). * Updated GMST observations, quoted from Sect. 6 of this update, are marked with an**
**asterisk, with "***very likely***" ranges given in brackets. ** In AR6 WGI, best-estimate values were not provided for warming**
**attributable to well-mixed greenhouse gases, other human forcings and natural forcings (though they did receive a "***likely***"**
**range); for comparison, best estimates (marked with two asterisks) have been retrospectively calculated in an identical way**
**to the best estimate that AR6 provided for anthropogenic warming (see discussion in Supplement Sect. S7.4.1). *** The**
**SR1.5 assessment drew only on GWI rounded to 0.1°C precision, whereas the repeat and updated calculations use the**
**updated multi-method assessment approach.**

| Estimates of warming attributable to multiple influences, in °C, relative to the 1850–1900 baseline period — Results are given as best estimates, with the *likely* range in brackets, and reported as Global Mean Surface Temperature (GMST). | | | | | |
|---|---|---|---|---|---|
| Definition ➡ | **(a) IPCC AR6 Attributable Warming Update** *Average value for previous 10-year period* | | | **(b) IPCC SR1.5 Attributable Warming Update** *Value for single-year period* | |
| Period ➡ Component ⬇ | **(i) 2010-2019** *Quoted from AR6 Chapter 3 Sect. 3.3.1.1.2 Table 3.1* | **(ii) 2010-2019** *Repeat calculation using the updated methods and datasets* | **(iii) 2014-2023** *Updated value using updated methods and datasets* | **(i) 2017** *Quoted from SR1.5 Chapter 1 Sect. 1.2.1.3* | **(ii) 2017** *Repeat calculation using the updated methods and datasets* | **(iii) 2023** *Updated value using updated methods and datasets* |
| **Observed** | 1.06 (0.88 to 1.21) | 1.07 (0.89 to 1.22) * | 1.19 (1.06 to 1.30) * | - | - | 1.43 (1.32 to 1.53) |
| **Anthropogenic** | 1.07 (0.8 to 1.3) | 1.09 (0.9 to 1.3) | 1.19 (1.0 to 1.4) | 1.0 (0.8 to 1.2) *** | 1.15 (0.9 to 1.4) | 1.31 (1.1 to 1.7) |
| **Well-mixed greenhouse gases** | 1.40** (1.0 to 2.0) | 1.38 (1.0 to 1.8) | 1.47 (1.0 to 1.9) | N/A | 1.43 (1.0 to 1.9) | 1.57 (1.1 to 2.1) |
| **Other human forcings** | -0.32** (-0.8 to 0.0) | -0.28 (-0.7 to 0.1) | -0.27 (-0.7 to 0.1) | N/A | -0.28 (-0.7 to 0.1) | -0.26 (-0.7 to 0.1) |
| **Natural forcings** | 0.03** (-0.1 to 0.1) | 0.05 (-0.1 to 0.2) | 0.04 (-0.1 to 0.2) | N/A | 0.04 (-0.1 to 0.2) | 0.04 (-0.1 to 0.2) |


The repeat calculations for attributable warming in 2010–2019 exhibit good correspondence with the results in AR6
WGI for the same period (see also Supplement, Sect. S7). The repeat calculation for the level of attributable
anthropogenic warming in 2017 is about 0.1 °C larger than the estimate provided in SR1.5 for the same period,





resulting from changes in methods and observational data (see AR6 WGI Chapter 2 Box 2.3). The updated results for
warming contributions in 2023 are higher than in 2017 due also to 6 additional years of increasing anthropogenic
forcing. Note also that the SR1.5 assessment only used the GWI method, whereas these annual updates apply the full
AR6 multi-method assessment (see Supplement Sect. S7.4 for details and rationale). A repeat assessment using the
SR1.5 trend-based definition (see Sect. 7.1) leads to results that are very similar to the single-year results reported in
Table 6b; best estimates across all components for single-year and trend-based definitions are identical to each other
for 2023, and identical or well within uncertainty range for 2017 (Supplement, Sect. S7.3 Table S3).

In this 2024 update, we assess the 2014–2023 decade average human induced-warming at 1.19 [1.0 to 1.4] °C, which
is 0.12°C above the AR6 assessment for 2010–2019. The single year average human-induced warming is assessed to
be 1.31 [1.1 to 1.7] °C in 2023 relative to 1850–1900. This best-estimate for the current level of human-induced
warming reaches the 1.3°C threshold for the first time. The best estimate is below the observed temperature in 2023
(1.43 [1.32 to 1.53] °C, see Sect. 6), but note the overlap of uncertainties. These best estimates for decade-average and
single-year human-induced warming are both 0.05 °C above the value estimated in the previous update for the year
2022 (Forster et al., 2023) – a rise partly driven by the high temperatures observed in 2023. Comparing our estimates
of attributable warming in 2017 with those reported last year by Forster et al. (2023), we find that attribution methods
give a slightly stronger anthropogenic warming, driven by the inclusion of observations for 2023. This is comprised
of a larger greenhouse gas attributable warming, partially offset by a slightly stronger aerosol-induced cooling, WGI
AR6 found that, averaged for the 2010–2019 period, essentially all observed global surface temperature change was
human-induced, with solar and volcanic drivers and internal climate variability making a negligible contribution. This
conclusion remains the same for the 2014–2023 period. Generally, whatever methodology is used, on a global scale,
the best estimate of the human-induced warming is (within small uncertainties) similar to the observed global surface
temperature change (Table 6).

### 7.4 Rate of human-induced global warming


Estimates of the human-induced warming rate refer to the rate of increase in the level of attributed anthropogenic
warming over time; this is distinct from the rate of increase in the observed global surface temperature (Sect. 6) which
is affected by internal variability such as El Niño and natural forcings such as volcanic activity (Jenkins et al 2023).
The rate of anthropogenic warming is driven by the rate of change of anthropogenic ERF, meaning variations in the
rate of climate forcing over time correlate with variations in the rate of attributed warming (see Fig. 8).

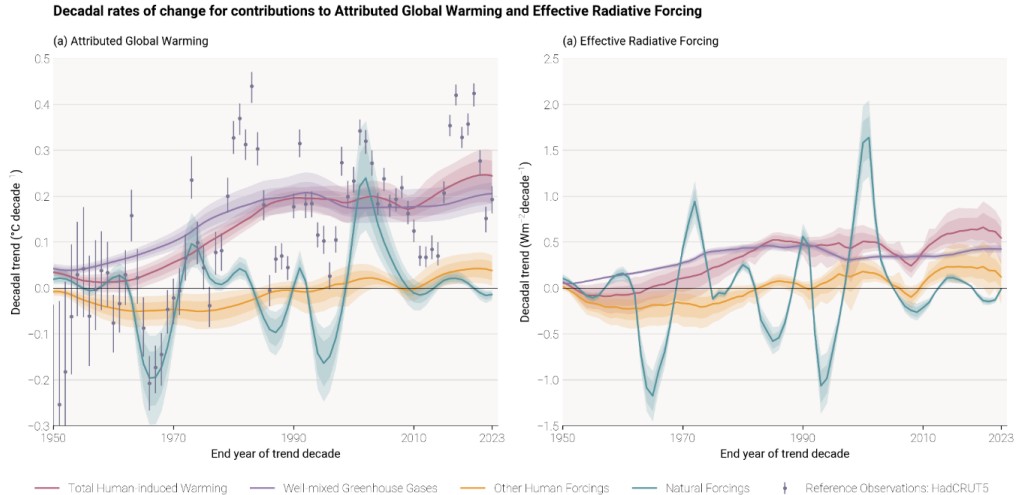


**Figure 8 Rates of (a) attributable warming (global mean surface temperature (GMST)) and (b) effective radiative forcing. The attributable warming rate time-series are calculated using the Global Warming Index method with full ensemble uncertainty. The observed GMST rates included for reference are also calculated with uncertainty from the HadCRUT5 ensemble, and, for consistency with the attributed warming rates, do not include standard regression error, which, for observed warming, would increase the size of the error bars. The effective radiative forcing rates are calculated using a representative 1000-member ensemble of the forcings provided in Sect. 4 of this paper.**


A very simple estimate of the rate of human-induced warming and effective radiative forcing was made last year by
Forster et al. 2023, which indicated that warming rates were unprecedented, surpassing 0.2 °C per decade (although
no uncertainty range was given). That rate calculation was based on annual changes in decade-average anthropogenic
warming levels from the GWI method (see Supplement Sect. S7.2). This year, attributed anthropogenic warming rates
are calculated for all attribution methods using linear trends, as used in AR6, with the overall rate estimate updated in
a manner that is fully traceable to and consistent with the rate assessment in AR6.

**7.4.1 SR1.5 and AR6 definitions of warming rate**
In recent IPCC assessments the definition of warming rate follows two approaches, both of which rely to some extent
on expert judgment. In SR1.5 several studies were considered, each defining the rate of warming in various ways and
over various timescales; the assessment concluded that the rate of increase of anthropogenic warming in 2017 was
0.2°C per decade with a likely range of 0.1°C to 0.3°C per decade). In AR6 WGI the rate of anthropogenic warming
utilised three methods (GWI, KCC, ROF, see Supplement Sect. S7.2) with the rate defined consistently across all
three as the linear trend in the preceding decade of attributed anthropogenic warming. While the best estimate trends
reported in AR6 were all higher than the SR1.5 assessment, Eyring et al. (2021) concluded that there was insufficient
evidence to change the SR1.5 assessed anthropogenic warming trend in the AR6 WGI report, which therefore



remained unchanged from SR1.5 at 0.2°C per decade (with a likely range of 0.1°C to 0.3°C per decade). Both the
SR1.5 and AR6 assessments were given to 0.1°C per decade precision only.

**7.4.2 Methods**

Following AR6's definition, the rate of warming is defined here as the rolling 10-year linear trend in attributed
anthropogenic warming, calculated using ordinary least-squares linear regression. Note that, as with the level of
anthropogenic warming, this decadal approach means the rate of warming in a given year is the trend centered on the
preceding decade (i.e. it is 5 years out of date). Each of the three attribution methods used to calculate the level of
warming are again used here to estimate separate anthropogenic warming rates.

Note that only the GWI methodology relies on the updated historical forcing timeseries presented in Sect. 4, with the
other two methods (ROF and KCC) relying on CMIP6 SSP2-4.5 simulations, which are increasingly out of date (see
Supplement Sect. S7.2). Very recent changes in anthropogenic forcing, for example desulphurisation of shipping fuels
or the impact of COVID-19, may therefore not be captured fully in the decade-average trend. Further, the
anthropogenic forcing record used for attributing warming contains small contributions from biomass burning in the
natural environment, because of difficulty separating this in estimates of anthropogenic aerosol emissions. It is not
expected that either of these effects substantially bias the globally-averaged rate of warming estimated here.

**7.4.3 Results**

Estimates from the GWI (based on observed warming and forcing), and KCC (based on CMIP simulations), both
report results in terms of GMST and are in close agreement across each time period. Estimates derived with the the
ROF method (also based on CMIP simulations), are also reported for GMST here and are more strongly influenced
by residual internal variability that remains in the anthropogenic warming signal due to the limitations in size of the
CMIP ensemble, as reflected in their broader uncertainty ranges. Given that the ROF results are in this sense outlying,
the standard approach of taking the median result for the overall multi-method assessment is adopted.

Results for human-induced warming rate are summarised in Table 7 and Fig 8. For the purpose of providing annual
updates, we take the median estimate at 0.01°C/decade precision, resulting in an overall best estimate for 2014–2023
of 0.26°C/decade. This increased rate relative to the 0.2°C/decade AR6 assessment is broken down in the following
way: (i) 0.03°C/decade of the increase is from a change in rounding precision (updating the AR6 assessment for the
2010–2019 warming rate from 0.2°C/decade to 0.23°C/decade), (ii) 0.02°C/decade of the increase is due to
methodological and dataset updates (updating the 2010–2019 warming rate from 0.23°C/decade to 0.25°C/decade;
this includes the effect of adding 4 additional observed years which affect the attribution for the entire historical
period), and (iii) only 0.01°C/decade of the increase is due to a substantive increase in rate for the 2014–2023 period
since the 2010–2019 period (updating 0.25°C/decade for 2010–2019 to 0.26°C/decade for 2014–2023). The spread of
rates across the three attribution methods remains similar to their spread in AR6, and hence do not support a decrease



in the uncertainty width in this update. However, to better reflect the closer agreement of the 5% floors and the larger
spread in the 95% ceilings of the three methods, and high rate from the ROF method, we update the uncertainty range
for the rate of human-induced warming from [0.1–0.3]°C/decade in AR6 to [0.2–0.4]°C/decade, leaving the precision
and width unchanged, noting that this is asymmetric around the central estimate. Therefore, the rate of human-induced
warming for the 2014-2023 decade is concluded to be 0.26°C/decade with a range of [0.2–0.4]°C/decade).

**Table 7 Updates to the IPCC AR6 rate of human-induced warming. Results for each method are given as best estimates**
**with 5-95% confidence, as described in the main text; assessment results are given as a best estimate with *likely* range in**
**brackets. Results from AR6 WGI (Ch.3 Sect. 3.3.1.1.2 Table 3.1) are quoted in column (i), and compared with a repeat**
**calculation using the updated methods and datasets in column (ii), and finally updated for the 2014-2023 period in column**
**(iii). The AR6 assessment result was identical to the SR1.5 assessment result, though the latter was based on a different set**
**of studies and timeframes. * Note that for clarity and ease of comparison with this year's updated assessment, in the assessed**
**rate in column (i) both quotes the the assessment from AR6 and retrospectively applies the median approach adopted in**
**this paper.**

| Estimates of anthropogenic warming rate, in °C per decade Results are given as best estimates, with brackets giving the *likely* range for the assessments, and 5-95% uncertainty for the individual methods | | | |
|---|---|---|---|
| Definition ➡ | **IPCC AR6 Anthropogenic Warming Rate Update** *Linear trend in anthropogenic warming over the trailing 10-year period* | | |
| Period ➡ | **(i) 2010-2019** *Quoted from AR6 Chapter 3 Sect. 3.3.1.1.2 Table 3.1* | **(ii) 2010-2019** *Repeat calculation using the updated methods and datasets* | **(iii) 2014-2023** *Updated value using updated methods and datasets* |
| Method ⬇ | | | |
| **Anthropogenic Warming Rate Assessment** | Quoted from AR6: 0.2 (0.1 to 0.3)  Using the median approach: 0.23 (0.1 to 0.3) * | 0.25 (0.2 to 0.4) | 0.26 (0.2 to 0.4) |
| GWI | 0.23 (0.19 to 0.35) GMST | 0.24 (0.18 to 0.29) GMST | 0.25 (0.19 to 0.30) GMST |
| KCC | 0.23 (0.18 to 0.29) GSAT | 0.25 (0.23 to 0.30) GMST | 0.26 (0.20 to 0.31) GMST |
| ROF | 0.35 (0.30 to 0.41) GSAT | 0.27 (0.17 to 0.38) GMST | 0.38 (0.24 to 0.52) GMST |


Fig. 8 and Table 7 include a breakdown into well-mixed GHGs and other human forcings (including aerosols), and
natural forcing contributions since pre-industrial times. The rate timeseries with ensemble uncertainty are depicted
from the GWI method, which is based on observed warming and historical forcing. The rate of total attributable
warming (the sum of anthropogenic and natural, not plotted) has good correspondence with the reference plotted
observed warming rates. The rates for the attributed warming also correlate closely with the forcing rates. Warming



rates have remained high due to strong GHG warming from high emissions and declining aerosol cooling (Forster et
al., 2023, Quaas et al.,2022, Jenkins et al., 2022).
**8 Remaining Carbon Budget**
AR5 (IPCC, 2013) assessed that global surface temperature increase is close to linearly proportional to the total
amount of cumulative $CO_2$ emissions (Collins et al., 2013). The most recent AR6 report reaffirmed this assessment
(Canadell et al., 2021). This near-linear relationship implies that for keeping global warming below a specified
temperature level, one can estimate the total amount of $CO_2$ that can ever be emitted. When expressed relative to a
recent reference period, this is referred to as the remaining carbon budget (Rogelj et al., 2018).

AR6 assessed the remaining carbon budget (RCB) in Chap. 5 of its WGI report (Canadell et al., 2021) for 1.5, 1.7 and
2 °C thresholds (see Table 7). They were also reported in its Summary for Policymakers (Table SPM.2, IPCC, 2021b).
These are updated in this section using the same method as last year (Forster et al., 2023).

The RCB is estimated by application of the WGI AR6 method described in Rogelj et al. (2019), which involves the
combination of the assessment of five factors: (i) the most recent decade of human-induced warming (given in Sect.
7), (ii) the transient climate response to cumulative emissions of $CO_2$ (TCRE), (iii) the zero emissions commitment
(ZEC), (iv) the temperature contribution of non-$CO_2$ emissions and (v) an adjustment term for Earth system feedbacks
that are otherwise not captured through the other factors. AR6 WGI reassessed all five terms (Canadell et al., 2021).
The incorporation of factor (v) was further considered by Lamboll and Rogelj (2022). Lamboll et al. (2023) further
considered factor (iv).

The RCB for 1.5, 1.7 and 2 °C warming levels is re-assessed based on the most recent available data. Estimated RCBs
are reported in Table 8. They are expressed both relative to 2020 to compare to AR6 and relative to the start of 2024
for estimates based on the 2014–2023 human-induced warming update (Sect. 7). Note that between the start of 2020
and the end of 2023, about 164 $GtCO_2$ has been emitted (Sect. 2). Based on the variation in non-$CO_2$ emissions across
the scenarios in AR6 WGIII scenario database, the estimated RCB values can be higher or lower by around 200 $GtCO_2$
depending on how deeply non-$CO_2$ emissions are reduced (Lamboll et al., 2023). The impact of non-$CO_2$ emissions
on warming includes both the warming effects of other greenhouse gases such as methane and the cooling effects of
aerosols such as sulfates. Updating these pathways increased the estimate of the importance of aerosols, which are
expected to decline with time in low emissions pathways (Rogelj et al., 2014), causing a warming and decreasing the
RCB (Lamboll et al., 2023). The AR6 WGIII version of MAGICC is used here. Structural uncertainties give inherent
limits to the precision with which remaining carbon budgets can be quantified. These particularly impact the 1.5 °C
RCB. Overall, the 1.5 °C compatible budget is very small and shrinking fast due to continuing high global $CO_2$
emissions.



**Table 8 Updated estimates of the remaining carbon budget for 1.5, 1.7 and 2.0 °C, for five levels of likelihood, considering**
**only uncertainty in TCRE. Estimates start from AR6 WGI estimates (first row for each warming level), updated with the**
**latest MAGICC emulator and scenario information from AR6 WGIII (from second row for each warming level), and an**
**update of the anthropogenic historical warming, which is estimated for the 2014–2023 period (third row for each warming**
**level). Estimates are expressed relative to either the start of the year 2020 or 2024. The probability includes only the**
**uncertainty in how the Earth immediately responds to carbon emissions, not long-term committed warming or uncertainty**
**in other emissions. All values are rounded to the nearest 50 GtCO$_2$.**

| Remaining carbon budget case/update | Base year | Estimated remaining carbon budgets from the beginning of base year (GtCO$_2$) | | | | |
|---|---|---|---|---|---|---|
| Likelihood of limiting global warming to temperature limit | | 17% | 33% | 50% | 67% | 83% |
| 1.5 °C from AR6 WG1 | 2020 | 900 | 650 | 500 | 400 | 300 |
| + AR6 emulators and scenarios | 2020 | 750 | 500 | 400 | 300 | 200 |
| **+ Updated warming estimate** | **2024** | **400** | **250** | **150** | **100** | **50** |
| 1.7 °C from AR6 WG1 | 2020 | 1450 | 1050 | 850 | 700 | 550 |
| + AR6 emulators and scenarios | 2020 | 1300 | 950 | 750 | 600 | 500 |
| **+ Updated warming estimate** | **2024** | **950** | **700** | **550** | **400** | **300** |
| 2 °C from AR6 WG1 | 2020 | 2300 | 1700 | 1350 | 1150 | 900 |
| + AR6 emulators and scenarios | 2020 | 2200 | 1650 | 1300 | 1100 | 900 |
| **+ Updated warming estimate** | **2024** | **1850** | **1350** | **1100** | **900** | **700** |


Updated RCB estimates presented in Table 8 for 1.5, 1.7 and 2.0 °C of global warming are smaller than AR6, and
geophysical and other uncertainties therefore have become larger in relative terms. This is a feature that will have to
be kept in mind when communicating budgets. The estimates presented here differ from those presented in the annual
Global Carbon Budget (GCB) publications (Friedlingstein et al., 2023). The GCB 2023 used the average between the
AR6 WGI estimate and the Forster et al. (2023) estimates. The RCB estimates presented here consider the same
updates in historical CO$_2$ emissions from the GCB as well as the latest available quantification of human-induced
warming to date and a reassessment from AR6 of non-CO$_2$ warming contributions.

The RCB for limiting warming to 1.5 °C is rapidly diminishing. It is important, however, to correctly interpret this
information. RCB estimates consider projected reductions in non-CO$_2$ emissions that are aligned with a global
transition to net zero CO$_2$ emissions (Lamboll et al., 2023). These estimates assume median reductions in non-CO$_2$





emissions between 2020–2050 of $CH_4$ (50 %), $N_2O$ (25 %) and $SO_2$ (77 %). If these non-$CO_2$ greenhouse gas emission
reductions are not achieved, the RCB will be smaller (see Lamboll et al., 2023 and Supplement, Sect. S8). Note that
the 50 % RCB is expected to be exhausted a few years before the 1.5 °C global warming level is reached due to the
way it factors future warming from non-$CO_2$ emissions into its estimate.
**9 Climate and weather extremes**
Changes in climate and weather extremes are among the most visible effects of human-induced climate change. Within
AR6 WGI, a full chapter was dedicated to the assessment of past and projected changes in extremes on continents
(Seneviratne et al., 2021), and the chapter on ocean, cryosphere and sea level changes also provided assessments on
changes in marine heatwaves (Fox-Kemper et al., 2021). Global indicators related to climate extremes include
averaged changes in climate extremes, for example, the mean increase of annual minimum and maximum temperatures
on land (AR6 WGI Chap. 11, Fig. 11.2, Seneviratne et al., 2021) or the area affected by certain types of extremes
(AR6 WGI Chap. 11, Box 11.1, Fig. 1, Seneviratne et al., 2021; Sippel et al., 2015). In contrast to global surface
temperature, extreme indicators are less established. Land average annual maximum temperature (TXx).

The climate indicator of changes in temperature extremes consists of land average annual maximum temperatures
(TXx) (excluding Antarctica). As part of this update, we provide an upgraded version of Fig. 6 from Forster et al.
(2023), which in turn is based on Fig. 11.2 from Seneviratne et al. (2021) (Fig. 9). As last year, three datasets are
analyzed: HadEX3 (Dunn et al., 2020), Berkeley Earth Surface Temperature (building off Rohde et al., 2013), and the
fifth-generation ECMWF atmospheric reanalysis of the global climate (ERA5; Hersbach et al., 2020). HadEX3 is
currently static and is not being updated. Berkeley Earth has been updated, resulting in TXx differences for most years
(less than 0.1°C), and now includes data for 2022. Of the three datasets, only ERA5 covers the whole of 2023 at the
present time. TXx is calculated by averaging the annual maximum temperature over all available land grid points
(excluding Antarctica) and then converted to anomalies with respect to a base period of 1961–1990. To express the
TXx as anomalies with respect to 1850–1900, we add an offset of 0.52°C to all three datasets. See Supplement Sect.
S9 for details on the data selection, averaging and offset computation.

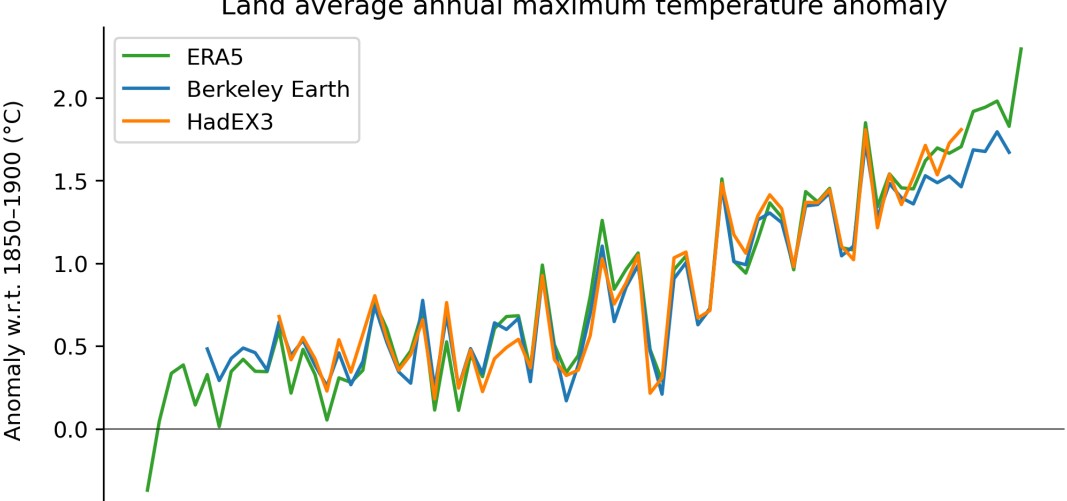


**Figure 9 Time series of observed temperature anomalies for land average annual maximum temperature (TXx) for ERA5**
**(1950–2023), Berkeley Earth (1955–2022) and HadEX3 (1961–2018), with respect to 1850–1900. Note that the datasets have**
**different spatial coverage and are not coverage-matched. All anomalies are calculated relative to 1961–1990, and an offset**
**of 0.52 °C is added to obtain TXx values relative to 1850–1900. Note that while the HadEX3 numbers are the same as shown**
**in Seneviratne et al. (2021) Fig. 11.2, these numbers were not specifically assessed.**

Our climate has warmed rapidly in the last few decades (Sect. 6), which also manifests in changes in the occurrence
and intensity of climate and weather extremes. From about 1980 onwards, all employed datasets point to a strong TXx
increase, which coincides with the transition from global dimming, associated with aerosol increases, to brightening,
associated with aerosol decreases (Wild et al., 2005, Sect. 3). The ERA5 based TXx warming estimate w.r.t. 1850-
1900 for 2023 is at 2.3°C; an increase of more than 0.5°C compared to 2022, and shattering the previous record by
more than 0.3°C. On longer time scales, land average annual maximum temperatures have warmed by more than
0.6 °C in the past 10 years (1.81 °C with respect to pre-industrial conditions) compared to the first decade of the
millennium (1.21°C; Table 9). Since the offset relative to our pre-industrial baseline period is calculated over the
1961–1990, temperature anomalies align by construction over this period but can diverge afterwards. In an extensive
comparison of climate extreme indices across several reanalyses and observational products, Dunn et al. (2022) point
to an overall strong correspondence between temperature extreme indices across reanalysis and observational
products, with ERA5 exhibiting especially high correlations to HadEX3 among all regularly updated datasets.

**Table 9 Anomalies of land average annual maximum temperature (TXx) for recent decades based on HadEX3 and ERA5.**

| Period | Anomaly w.r.t. 1850-1900 (°C) | Anomaly w.r.t. 1961-1990 (°C) | Anomaly w.r.t. 1961-1990 (°C) |
|--------|-------------------------------|-------------------------------|-------------------------------|





|  | ERA5 | ERA5 | HadEX3 |
|---|---|---|---|
| 2000-2009 | 1.21 | 0.69 | 0.72 |
| 2009-2018 | 1.54 | 1.02 | 1.01 |
| 2010-2019 | 1.62 | 1.11 | - |
| 2011-2020 | 1.63 | 1.12 | - |
| 2012-2021 | 1.70 | 1.18 | - |
| 2013-2022 | 1.73 | 1.21 | - |
| 2014-2023 | 1.81 | 1.29 | - |

**10 Code and data availability**
The main indicators are presented in an online dashboard at climatechangetracker.org,
https://climatechangetracker.org/igcc.

The carbon budget calculation is available from https://github.com/Rlamboll/AR6CarbonBudgetCalc/tree/v1.0.1
(Lamboll and Rogelj, 2024). The code and data used to produce other indicators are available in repositories under
https://github.com/ClimateIndicator/data/tree/v2024.04.25 (Smith et al., 2024b). All data are available from
https://doi.org/10.5281/zenodo.11064126 (Smith et al., 2024a). Data are provided under the CC-BY 4.0 Licence.

HadEX3 [3.0.4] data were obtained from https://catalogue.ceda.ac.uk/uuid/115d5e4ebf7148ec941423ec86fa9f26
(Dunn et al., 2023) on 5 April 2023 and are © British Crown Copyright, Met Office, 2022, provided under an Open
Government Licence; http://www.nationalarchives.gov.uk/doc/open-government-licence/version/2/ (last access: 2
June 2023).
**11 Discussion and conclusions**
The second year of the Global Climate Change (IGCC) initiative has built on last year's effort and the AR6 report
cycle to provide a comprehensive update of the climate change indicators required to estimate the human-induced
warming and the remaining carbon budget. Table 10 presents a summary of the headline indicators from each section
compared to those given in the AR6 assessment and also summarises methodological updates. The main substantive
dataset change since AR6 is that land-use $CO_2$ emissions have been revised down by around 2 $GtCO_2$ (Table 10).
However, as $CO_2$ ERF and human-induced warming estimates depend on concentrations, not emissions, this does not
affect most of the other findings. Note it does slightly increase the remaining carbon budget, but this is only by
5 $GtCO_2$, less than the 50 $GtCO_2$ rounding precision.

**Table 10 Summary of headline results and methodological updates from the Indicators of Global Climate Change (IGCC)**
**initiative.**



| Climate Indicator | AR6 2021 assessment | This 2023 assessment | Explanation of changes | Methodological updates since AR6 |
|---|---|---|---|---|
| Greenhouse gas emissions<br><br>AR6 WGIII Chapter 2: Dhakal et al. (2022); see also Minx et al. (2021) | 2010-2019 average:<br><br>56 ± 6 GtCO$_2$e* | 2010-2019 average:<br><br>53 ± 5.5 GtCO$_2$e<br><br>2013-2022 average:<br><br>54 ± 5.4 GtCO$_2$e | Average emissions in the past decade grew at a slower rate than in the previous decade. The change from AR6 is due to a systematic downward revision in CO$_2$-LULUCF and CH$_4$ estimates. Real-world emissions have slightly increased. | CO$_2$-LULUCF emissions revised down. PRIMAP-hist CR used in place of EDGAR for CH$_4$ and N$_2$O emissions, atmospheric measurements taken for F-gas emissions. These changes reduce estimates by around 3 GtCO$_2$e (Sect. 2). Note following convention, ODS F-gases are excluded from the total. |
| Greenhouse gas concentrations<br><br>AR6 WGI Chapter 2: Gulev et al. (2021) | 2019:<br><br>CO$_2$, 410.1 [± 0.36] ppm<br><br>CH$_4$, 1866.3 [± 3.2] ppb<br><br>N$_2$O, 332.1 [± 0.7] ppb | 2022:<br><br>CO$_2$, 419.2 [±0.4] ppm<br><br>CH$_4$, 1922.9 [±3.3] ppb<br><br>N$_2$O, 337.0 [±0.4] ppb | Increases caused by continued GHG anthropogenic emissions | Updates based on NOAA data and AGAGE (Sect. 3) |
| Effective radiative forcing change since 1750<br><br>AR6 WGI Chapter 7: Forster et al. (2021) | 2019:<br><br>2.72 [1.96 to 3.48] W m$^{-2}$ | 2023:<br><br>2.79 [1.78 to 3.60] W m$^{-2}$ | Trend since 2019 is caused by increases in greenhouse gas concentrations and reductions in aerosol precursors. Shipping emission reductions may have added approximately 0.1 W m$^{-2}$ to the ERF in 2023 compared to 2022. However, increases in biomass burning aerosol from Canadian wildfires decreased the ERF by more. | Follows AR6 with minor update to aerosol precursor treatment that does not affect historic estimates |



| | | | | |
|---|---|---|---|---|
| **Earth's energy imbalance**<br><br>**AR6 WGI Chapter 7: Forster et al. (2021)** | **2006-2018 average:**<br><br>**0.79 [0.52 to 1.06] W m$^{-2}$** | **2010-2023. average:**<br><br>**0.96 [0.67 to 1.26] W m$^{-2}$** | **Substantial increase in energy imbalance estimated based on increased rate of ocean heating.** | **Ocean heat content timeseries extended from 2018 to 2023 using 4 of the 5 AR6 datasets. Other heat inventory terms updated following von Schuckmann et al (2023a). Ocean heat content uncertainty is used as a proxy for total uncertainty. Further details in Sect. 5.** |
| **Global mean surface temperature change above 1850-1900**<br><br>**AR6 WGI Chapter 2: Gulev et al. (2021)** | **2011-2020 average:**<br><br>**1.09 [0.95 to 1.20] °C** | **2014-2023 average:**<br><br>**1.19 [1.06–1.30] °C** | **An increase of 0.1 °C within three years, indicating a high decadal rate of change which may in part be internal variability.** | **Methods match four datasets used AR6 (Sect. 6). Individual datasets have updated historical data, but these changes are not materially affecting results.** |
| **Human induced global warming since preindustrial**<br><br>**AR6 WGI Chapter 3: Eyring et al. (2021)** | **2010-2019 average:**<br><br>**1.07 [0.8 to 1.3] °C** | **2010-2019 average:**<br><br>**1.09 [0.9 to 1.3] °C**<br><br>**2014-2023 average:**<br><br>**1.19 [1.0 to 1.4] °C** | **An increase of 0.1 °C within four years, indicating a high decadal rate of change. GMST increase in 2023 has revised historical estimates upwards.** | **The three methods for the basis of the AR6 assessment are retained, but each has new input data (Sect. 7)** |
| **Remaining carbon budget for 50% likelihood of limiting global warming to 1.5°C**<br><br>**AR6 WGI Chapter 5: Canadell et al. (2021)** | **From the start of 2020:**<br><br>**500 GtCO$_2$** | **From the start of 2024:**<br><br>**150 GtCO$_2$** | **The 1.5°C budget is becoming very small. The RCB can exhaust before the 1.5°C threshold is reached due to having to allow for future non-CO$_2$ warming.** | **Emulator and scenario change has reduced budget since 2020 by 100 GtCO$_2$ (Sect. 8)** |
| **Land average maximum temperature change compared to pre-industrial.**<br><br>**AR6 WGI Chapter 11:** | **2009-2018 average:**<br><br>**1.55 °C** | **2014-2023 average:**<br><br>**1.74 °C** | **Rising at a substantially faster rate compared to global mean surface temperature** | **HadEX3 data used in AR6 replaced with reanalysis data employed in this report which is more updatable going forward. Adds 0.01 °C to estimate (Sect. 9)** |



| Seneviratne et al., 2021 | | | | |
|---|---|---|---|---|
| | | | | |


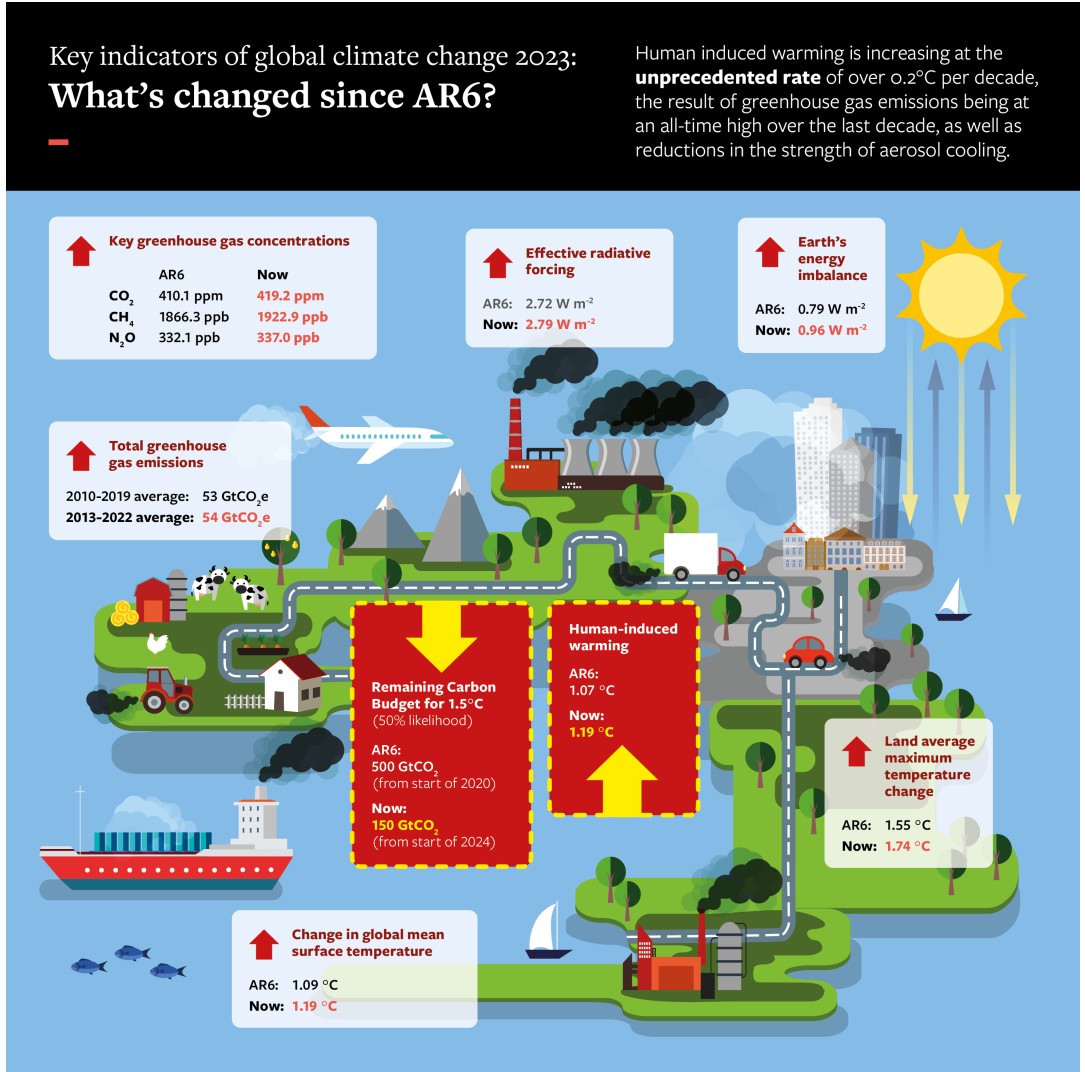




**Figure 10 Infographic for the best estimate of headlinee indicators assessed in this paper.**
Last year witnessed a large increase in GMST (Sect. 6), approaching 1.5°C above 1850-1900 levels that has widely
been reported in the press. The 2022 to 2023 increase was the third largest annual increase in the instrumental record
after 1876-1877 and 1976-1977, two other periods with a strong transition from La Niña to El Niño conditions. The
reasons for the change, especially regarding the potential role of external forcings such as shipping emission reductions
compared to internal variability are currently being investigated (e.g. Schmidt, 2024; Gettelman et al. 2024). Our work
looks at long-term changes and does not directly investigate the reasons for the jump in GMST levels, yet we note
that our best estimate of human induced warming in 2023 is 1.31 (1.1 to 1.7) °C (Table 6), below the observed GMST
estimate of 1.43 [1.32 to 1.53] °C in 2023 (Sect. 6). This indicates a potentially large role for El Niño and other wind-
driven ocean changes.
Methane and biomass emissions had a strong component of change related to climate feedbacks (Sects. 2 and 3). Such
changes will become increasingly important over this century, even if the direct human influence declines. These
changes need to be properly accounted for to explain atmospheric concentration and energy budget changes. The
approach to methane taken in this paper (where changes to natural sources are excluded) is inconsistent with that taken
for aerosol emissions (where wildfire changes are included). In future years and in the next IPCC report a consistent
approach to attribution of atmospheric emissions, concentration change and radiative forcing should be developed.
It is hoped that this update can support the science community in its collection and provision of reliable and timely
global climate data. In future years we are particularly interested in improving SLCF updating methods to get a more
accurate estimate of short-term ERF changes. The work also highlights the importance of high-quality metadata to
document changes in methodological approaches over time. In future years we hope to improve the robustness of the
indicators presented here but also extend the breadth of indicators reported through coordinated research activities.
For example, we could begin to make use of new satellite and ground-based data for better greenhouse monitoring
(e.g. via the WMO Global Greenhouse Gas Watch initiative). Parallel efforts could explore how we might update
indicators of regional climate extremes and their attribution, which are particularly relevant for supporting actions on
adaptation and loss and damage.

Generally, scientists and scientific organisations have an important role as "watchdogs" to critically inform evidence-
based decision-making. This annual update traced to IPCC methods can provide a reliable, timely source of
trustworthy information. As well as helping inform decisions, we can use the update to track changes in dataset
homogeneity between their use in one IPCC report and the next. We can also provide information and testing to
motivate updates in methods that future IPCC reports might choose to employ.

This is a critical decade: human-induced global warming rates are at their highest historical level, and 1.5 °C global
warming might be expected to be reached or exceeded within the next 10 years in the absence of cooling from major
volcanic eruptions (Lee et al., 2021). Yet this is also the decade that global greenhouse gas emissions could be expected
to peak and begin to substantially decline. The indicators of global climate change presented here show that the Earth's



energy imbalance has increased to around 0.9 W m$^{-2}$, averaged over the last 12 years. This also has implications for
the committed response of slow components in the climate system (glaciers, deep ocean, ice sheets) and committed
long-term sea level rise, but this is not part of the update here. However, rapid and stringent GHG emission decreases
such as those committed to at COP28 could halve warming rates over the next 20 years (McKenna et al., 2021). Table
1 shows that global GHG emissions are at a long-term high, yet there are signs that their rate of increase has slowed.
Depending on the societal choices made in this critical decade, a continued series of these annual updates could track
an improving trend for some of the the indicators herein discussed.

**Supplement**

The supplement related to this article is available online at: TBD

**Author contributions**

PMF, CJS, MA, PF, JR and AP developed the concept of an annual update in discussions with the wider IPCC
community over many years. CJS led the work of the data repositories. VMD, PZ, SS, JM, CFS, SIS, VN, AP, NG,
GP, BT, MSP, JR, PF, MA and PT provided important IPCC and UNFCCC framing. PMF coordinated the production
of the manuscript with support from DR. WFL led Sect. 2 with contributions from JM, PF, GP, JG, JP and RA. BH
led Sect. 3, CJS led Sect. 4 with contributions from BH, FD, SS, VN and XL. KvS and MDP led Sect. 5 with
contributions from LC, MI, TB and RK. BT led Sect. 6 with contributions from PT, CM, CK, JK, RR, RV and LC.
TW led Sect. 7 with contributions and calculations from AR, NG, SJ and MR. RL led Sect. 8 with contributions from
JR and KZ. Sect. 9 was led by MH, with contributions from SIS, XZ and DS. All authors either edited or commented
on the manuscript.

**Competing interests**

The contact author has declared that none of the authors has any competing interests.

**Disclaimer**

Publisher's note: Copernicus Publications remains neutral with regard to jurisdictional claims in published maps and
institutional affiliations.

**Acknowledgements**

This research has been supported by the European Union's Horizon Europe research and innovation programme under
Grant Agreement No. 820829, 101081395, 101081661 and 821003), the H2020 European Research Council (grant
no. 951542), the Natural Environment Research Council (NE/T009381/1) and the Engineering and Physical Research
Council (EP/V000772/1). Chris Smith, Matthew D. Palmer, Colin Morice, Rachel E. Killick and Richard A. Betts
were supported by the Met Office Hadley Centre Climate Programme funded by DSIT. Peter Thorne was supported
by Co-Centre award number 22/CC/11103. The Co-Centre award is managed by Science Foundation Ireland (SFI),
Northern Ireland's Department of Agriculture, Environment and Rural Affairs (DAERA) and UK Research and





Innovation (UKRI), and supported via UK's International Science Partnerships Fund (ISPF), and the Irish
Government's Shared Island initiative.

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
