# Peer review of "Indicators of Global Climate Change 2023: annual update of key"

_Earth System Science Data, 2024_

## Author Comment (AC1)

**Response to reviewers**

We thank the two reviewers and the four community comments for their encouraging and thoughtful insights. We respond to each in turn below.

**RC1: Matthew Jones**

Congratulations to the authors. This update on the inaugural report last year clearly represents a substantial effort, and the body evidence that it contains remains very valuable to a diverse sub-communities of Earth System Science.

My expertise connects more closely with some sections than others, and as such comments are concentrated on emissions estimates and fire fluxes. I trust that other reviewers will collectively have expertise across the board.

I think the authors have done an excellent job of explaining any new features / advances of the report, which is where I focussed my review.

**Thank you**

Below, I identify some considerations and suggest minor revisions to the text to add clarity where required (All line numbers are from tracked2024.pdf.):

L499-L506: Might be relevant to cite Minx et al. (2021) here.

**Now cited at the start of the paragraph.**

590-591: "The fossil fuel share of global greenhouse gas emissions was approximately 70% in 2022 (GWP100 weighted), based on the EDGAR v8 dataset (Crippa et al. 2023) and net land use CO2 emissions from the Global Carbon Budget." I had understood from earlier data source descriptions that the GCB data for CO2-FFI and CO2-LU and PRIMAP-hist CR for other gases was being preferred to EDGAR in this update. For clarity I would perhaps avoid presenting results based on EDGAR except for validation/uncertainty characterisation. Or did I mis-understand?

EDGAR is the only source with sufficient sector and gas level data to estimate this fraction (e.g. including the split of fossil vs non-fossil CH4 emissions in the energy sector), so was employed here.

842-844: "Such carbon cycle feedbacks are not considered here as they are not a direct emission from human activity, yet they will contribute to greenhouse gas concentration rise, forcing and energy budget changes discussed in the next sections. They will become more important to properly account for in future years." Is it possible to put an indicative number on the potential magnitude of influence?

**Without adding a lot more context and discussion of wildfires, permafrost thaw and dieback etc, this is hard to do so we have not quantified this here.**

I cannot see where the acronym "ODS" is defined. Likewise, I can't see where the acronym "IMO" is defined. As a general point, please could you check that all acronyms are defined.

ODS was defined on first use in section 2.1, IMO was only used once so is now spelt out, but we have been through the paper again to look for lost acronyms or where they can be replaced by full names.

I suggest using "GFED4s" throughout the manuscript. Also, is it definitely GFED4s and not GFED4.1s? The update (GFED5) is upcoming and will have significantly greater area burned and emissions, and so flagging the GFED version used in different publications has never been more important.

**GFED4.1s used – now clarified in two places**

170-173: "Whether human-caused burning, a feedback due to the extreme heat or naturally occurring, we choose to include them in our tracking, as historical biomass burning emissions inventories have previously been consistently treated as a forcing (for example in CMIP6), though this assumption may need to be revisited in the future". Do you mean "consistently treated as an *anthropogenic* forcing"? (If so and as an aside, that is quite an assumption! But I appreciate it's not really up to the authors to deal with that issue here).

Yes indeed, now added for clarity – we think because it has been hard to attribute.

169-170: You could mention the World Weather Attribution results showing impact of climate change on fire weather in Canada this year. This may be relevant too: https://essopenarchive.org/doi/full/10.22541/essoar.170914412.27504349

Good idea – Barnes et al. added

Barnes, C., Boulanger, Y., Keeping, T., Gachon, P., Gillett, N., Haas, O., Wang, X., Roberge, F., Kew, S., Heinrich, D., Singh, R., Vahlberg, M., Van Aalst, M., Otto, F., Kimutai, J., Boucher, J., Kasoar, M., Zachariah, M., and Krikken, F.: Climate change more than doubled the likelihood of extreme fire weather conditions in Eastern Canada, Imperial College London, https://doi.org/10.25561/105981, 2023.

Also, a couple of further points while on the 'fire' theme:

-- Is it relevant to your analyses that biomass burning emissions were extremely high in the northern extratropics but low in the African and South American savannahs? (i.e. Two of the regions with greatest contribution to mean annual). Basically, is the strange spatial distribution of short-lived forcers relevant on top of the global anomaly?

**Yes, but this is beyond the scope of this paper**

-- Should the relevance of fire be discussed in other sections beyond "Non-methane short-lived climate forcers" (I.e. for other gases)? My reading is that fire emissions are not included in the emissions data for CO2 / LLGHGs, except for those associated with LULUC. Presumably, the assumption is that any C lost is be recuperated as vegetation recovers, giving Net Zero on century-scale time horizons. If that is the case then it could be helpful to state it explicitly, given the enormous quantity of C emitted by Canadian fires (on the order of 0.5 GtC; https://atmosphere.copernicus.eu/copernicus-canada-produced-23-global-wildfire-carbon-emissions-2023) - equivalent to about 5 years of UK CO2 emissions. Recovery on the century horizon might be optimistic, especially when loss of old growth forest and peat / organic soils is considered (i.e. C dynamics are totally different to tropical grassland fires).

We now include the following sentence in section 2.1: "The GCB methodology includes CO2 emissions from deforestation and forest degradation fires, but excludes wildfires, which are assumed to be natural even if climate change affects their intensity and frequency."

Further, we already allude to this broader issue of climate feedbacks in the discussion section, and now expand it as follows: "Methane and biomass emissions had a strong component of change related to climate feedbacks (Sects. 2 and 3). Such changes will become increasingly important over this century, even if the direct human influence declines. These changes need to be properly accounted for to explain atmospheric concentration and energy budget changes. The approach to methane taken in this paper (where changes to natural sources are excluded) is inconsistent with that taken for aerosol emissions (where wildfire changes are included). In future years and in the next IPCC report a consistent approach to attribution of atmospheric emissions, concentration change and radiative forcing should be developed. Similarly, we follow the underlying literature in treating wildfire related CO2 emissions and removals as natural only (Friedlingstein et al. 2023), even though their intensity and frequency is shifting under anthropogenic climate change."

1676-1677: "We determine from provisional data that aviation activity in 2023 had not yet returned to pre-COVID levels" What is the provisional data?

We now reference IATA 2024

IATA: Air Passenger Monthly Analysis March 2024, https://www.iata.org/en/iata-repository/publications/economic-reports/air-passenger-market-analysis-march-2024/, accessed 20.05.2024, 2024.

2031-2036: Perhaps add a statement to clarify that human-induced warming is the dominant contributor to total warming here?

**This paragraph was confusing and unnecessary so it has now been deleted**

Section 7: I am curious how these results compare with our dataset "National contributions to climate change due to historical emissions of carbon dioxide, methane, and nitrous oxide since 1850" (paper: https://www.nature.com/articles/s41597-023-02041-1; latest data: https://zenodo.org/records/10839859). In that dataset, we apply IPCC-recommended TCRE and GWP(\*) constants to estimate national contributions to climate change (as well as the global total). Our input emissions data sources are similar to those used here (but NB our work included CO2, CH4, and N2O only). Can a comparison be made or is this challenging, e.g. due to incomplete overage of the forcers in our paper? Please don't feel obliged to do exhaustive work on this unless you think the comparison of your numbers with estimates based on "Emissions\*GWP\*TCRE" is useful in some way. (I am happy to remain curious.)

**Great idea, but this is future work 😊**

Figure 9: Is Berkeley Earth diverging from ERA5 after ~2015? If so, an explanation of possible reasons might be worthwhile.

**We don't know it looks like it might be but too early to say, we will look at this next year. A note has been added to the text**

2832-2835: The special note on HadEX seems a little out of place given the huge number of datasets that are adopted into this paper. Perhaps this should go in the acknowledgements or some other appropriate section? Or perhaps the data usage / licensing statements of lots of other datasets should also be given.

**This is a crown copyright dataset, so is highlighted as it has particular restrictions compared to the other datasets used**

Table 10: Cell in row "Greenhouse gas emissions", column "Methodological since AR6". My reading of the methods section was that GCB-FFI values had also replaced EDGAR values. Flagging this in case a clarification is needed.

**Table has now been corrected and methodological update given as suggested**

Table 10 is a really great summary of everything new in this report, with nice level of accessible detail, complementing a very comprehensive report.

**Thank you, it has been extended slightly to make methodological changes clearer.**

**RC2: David Huard**

**General comments**

This paper provides a valuable service to the scientific community. The cyclic nature of IPCC assessments often means expertise is lost and work is redone from scratch. Keeping the science "alive" and accessible between cycles is sure to make the work of the AR7 author teams smoother. One thing that worries me, however, are the opportunity costs. Are these updates diverting significant time and resources away from original research? Also, as other reviewers have pointed out, many more indicators could eventually be included in these updates. I'm wondering whether this model (one aggregating paper) will scale well to a broader set of indicators. These comments are not a criticism of the paper per say, but an invitation to work out a data publishing approach that reduces "maintenance" costs, spurs innovation, and improves the FAIRness of IPCC data.

Note that I'm not an expert in any of the indicators presented in the paper, so the review rather focuses on text and figure clarity.

Thank you for taking the time to review. Other team have shown an interest in both joining our effort to add other indicators and also in launching parallel efforts, so we have similar hopes to yourself.

**Technical corrections**

83-84: Should the number of decimal places for the central values and ranges match?

**Keeping the second decimal place was a specific choice to compare to the 1.43C**

84: I didn't understand this sentence (This is below...) on my first read. It only became clear after reading the section in the text distinguishing human induced warming with observed warming. I suggest replacing "record" by "warming".

**"Warming record" is now used for clarity**

106: Remove legend entry for "Available in Dashboard", since it's not used in the diagram. Also, it may not be clear to all readers what "Repo" stands for.

**Changes made as suggested – repository used**

323: typo (. .)

**Corrected**

498-499: I don't understand the distinction made between 0.15 W/m2/decade being a "constant pace", and 0.0.13 W/m2/decade being an "accelerated pace". I thought Figure 4 would make it clearer, but I get the impression that it's the ocean that's warming at an increasing rate. Please clarify.

**To clarify, we have modified the sentence to say**

"while the land, cryosphere, and atmosphere have been warming at a pace of  $0.013 \pm 0.003$  W m-2 per decade (Minière et al., 2023). "

523: In panel b, the IPCC AR6 results are not for "consecutive 20-year periods", and not in chronological order (1971-2006 is after 1971-2018). The results for "This study" clearly illustrate the increase in heating rate, but it's not obvious how to compare them to AR6 results. If the figure is meant to convey differences between AR6 and this study, please make the comparison easier for readers.

The methods have not changed since AR6, so the figure is only meant to show different periods. AR6 period data are the same in our assessment and in AR6. The caption is updated to make clear that there has been no change in AR6 period data estimates

582: From this section on, it feels like the syntax is less fluid.

We have adopted your suggestions below, thank you.

587-589: Review syntax, use punctuation.

Punctuation (a colon) has been added.

591-593: Review syntax

On reflection, the paragraph was unhelpful, so we have deleted it.

630: Single panel figures generally have no title. I note that there is a great diversity

across figure styles, label size and fonts, background color, etc. Using consistent styling would improve the cohesiveness of the paper.

Title removed. We agree on the need for consistency of figures and will look to improve this in future years.

664: Y-label reads bottom to top, but x-labels read top to bottom.

This figure style matches that used in IPCC AR6, so the style is retained.

688: In the text, uncertainty bounds are given with brackets [], while in the table they are enclosed in parentheses (). Also check Supplementary Material:376 for consistent use of brackets vs parentheses.

Changed to square brackets for uncertainty. In main paper, supplement and Figure 6.

765: typo (the the)

Deleted repeated word

781: I don't think "uncertainty width" is a standard expression. Just "uncertainty" would work. Later, "range" is used, so please use the same expression consistently.

"Range" now used, "width" changed in two places.

819-820: Please refer to the actual factors instead of their number ("factor (v)").

Now changed as suggested, this is much clearer, thank you

831: MAGICC is mentioned without explanation or a reference.

Sentence now deleted, it was not needed

901: I took a look at the Github repo and found it well organized and easy to navigate. I noticed however that units are not provided consistently across datasets. Some datasets include units in column headers, other in the accompanying yaml file, while some just don't specify units at all. Also, some of the repos that are referenced in the main README table lack their own README file.

This will be improved next year. Data was coming from different sources, we are steadily improving consistency.

925 Typo in Table 10: Global mean surface temperature change above 1850-1900, last column: (datasets used `in` AR6)

**edited**

956-957: Not sure I understand what is meant by "dataset homogeneity between their use in one IPCC report and the next".

Simplified to "track changes in datasets between their use in one IPCC report and the next."

**CC1: Ken Mankoff**

\*\*\* Title

I find the title overly broad (and uses the word "indicators" 2x). This manuscript ignores some key indicators of change (such as mass loss from ice sheets, sea level rise, and likely many others outside my research areas). In fact, L964/966 states, "This also has implications for the committed response of slow components in the climate system (glaciers, deep ocean, ice sheets) and committed long-term sea level rise, but this is not part of the update here." The abstract also suggests the focus of the manuscript is somewhat different from the title. L73-76 say "We compile monitoring datasets to produce estimates for key climate indicators related to forcing of the climate system: emissions of greenhouse gases and short-lived climate forcers, greenhouse gas concentrations, radiative forcing, the Earth's energy imbalance, surface temperature changes, warming attributed to human activities, the remaining carbon budget, and estimates of global temperature extremes."

From the abstract, forcings, emissions, GHG concentrations, budgets remaining, and attribution to humans are not "Indicators of Global Climate Change". They are "climate indicators" as per your Table 10. Surface temperature is an indicator of global climate change, as are temperature extremes. I suggest the title should better reflect the scope of the manuscript.

We note and thank you for your point and for reviewing the paper. We considered the title choice carefully but would like to keep it as is for now. We plan to add indicators over time: sea level rise and extremes will be included next year. There is also interest in parallel efforts. For now, we chose to keep the title consistent with its first year. We will revisit this choice going forward.

**\*\*\* Land change**

You highlight that land average maximum temperature change is "rising at a substantially faster rate compared to global mean surface temperature". If so, might

it also be useful, relevant, and important to highlight what temperatures are rising (or dropping) slower than the global mean surface temperature?

This land daily maximum is not part of the global mean estimate, which is based on monthly daily averages not daily maximum temperature data, so it is not straightforward to look into this.

L867: Is this a complete sentence?

This was an old redundant subtitle - now deleted

\*\*\* Figures

Fig 1: Why and how does the remaining budget depend on future emissions? I can see the remaining budget depending on future "scenarios" (a broader term) where scenario includes drawdown technologies.

As explained in Figure 1 and Section 8, it depends on future emissions of non-CO2 gases as assumptions need to be made about this as the budget is only for CO2.

Fig 3B: I cannot see which legend goes with the large negative forcing. I assume this is tropospheric aerosols, but am not certain. I become more certain when I see aerosols as negative in 3A. Perhaps labels should be next to lines at the far right side? There is also a dotted line but no dotted legend.

The dotted line is the zero line, the style matches IPCC AR6 figures, so we choose to retain the figure style. We think lines are readable, especially with comparing to Figure 3a, as you suggest.

Fig 4B: Are differences between AR6 and this study only due to different time periods, or also different methods? If different methods, can you do the same time periods as IPCC to show changes due only to methods and not time? Can the x-axis and display of this graphic be improved? For example, could the X axis span 1971 to 2023 and each of these could be box spanning the appropriate x-range, many of them overlapping?

The caption is extended to show that the data has not changed for the IPCC AR6 periods. The overlapping periods are now in the caption. We feel your suggestion might make things even more complex.

Infographic: top right text I suggest removing words "being" and "the strength of".

We think these words are important to clarify the ongoing nature of the GHG emissions and to avoid confusion on the sign of the aerosol change. We therefore

do not change the figure. Although note the carbon budget number has been revised to 200 GtCO2.

**CC2: 'Comment on essd-2024-149', Rasmus Benestad, 13 May 2024 reply**

This is a timely paper that brings up the discrepancy between the slow IPCC cycles and the fast rate of change that the real world undergoes. It would be tempting to say that the IPCC also needs to adapt to climate change and adjust how it works to match the needs for up-to-date information about the changing climate. However, it neglects an important part of the climate system: the global hydrological cycle.

Traditionally, data and climate indicators have emphasised temperature and warming but neglected the equally important aspect of disruption to the global hydrological cycle. While the global observation systems in the past were unable to provide reliable global status, modern satellite observations and reanalyses are becoming more mature. We have made the case for including global indicators for the hydrological part of Earth's climate system in https://doi.org/10.1007/s43832-024-00063-3 and https://doi.org/10.1371/journal.pclm.0000029 - there are indicators suggesting that the total amount of precipitation falling on Earth's surface each day has increased over the past and that the fractional surface area of the Earth receiving precipitation each day has shrunk. These are key to understanding changes in extreme rainfall and drought, and hence ought to be relevant in this paper (e.g. L97: "policymakers with annual updates of the latest scientific understanding on the state of selected critical indicators of the climate system and of human influence."). Preferably, resources such as https://pulse.climate.copernicus.eu/ should also display this kid of information too. Many of the headlines on natural disasters around the world refer to flooding and extreme rainfall (recently China, Brazil and Afghanistan), and Lussana et al. (https://doi.org/10.1002/joc.8375) reported that he hydrological cycle is undergoing changes in all regions, and a common features includes an increase in intense precipitation events and a decrease in the corresponding spatial extent.

We completely agree! We attempted to bring in some hydrological indicators this year, namely total land precipitation and also an extreme rainfall indicator. This is being led by groups in Korea and India. Unfortunately, they were not ready in time and this effort will need to wait until next year. This community are trying to establish a wider parallel effort.

**'Comment on essd-2024-149', Gareth S. Jones, 20 May 2024**

The authors have made an ambitious attempt to update several climate assessments made by the Intergovernmental Panel on Climate Change 6th working group 1 assessment.

Below are four comments aligned with 1) IPCC association, 2) FAIR principles, 3) Detection and Attribution method choices, and 4) Statistical interpretation - Aliasing.

Throughout I will refer to the Forster et al., 2023 study as F23 and the current study as F24.

**1) IPCC association**

The authors have closely aligned themselves with the Intergovernmental Panel on Climate Change (IPCC) assessments (e.g., Line 109).

However, IPCC assessments involve many more scientists from a wider geographic diversity than in this study, and assesses a wider range of scientific developments than done here.

In a couple of places the authors state that the aim is to track changes between IPCC reports (L114 and L956).

As we don't know who the authors of the next IPCC assessment (AR7) will be, what studies they will assess, let alone what conclusions they may come to, this is at best an optimistic aim.

The study should thus distance itself more from the IPCC process, and some of the authors promoting the work be more circumspect about the links to IPCC report cycles.

They would then be free to make appropriate choices about what methods and data to change, which may be more useful for policymakers and may be a more valuable contribution to a future AR7 assessment of the scientific literature.

We disagree with this judgement on our paper, we are clear that we are not trying to reproduce the IPCC report. E.g. lines 113-116 "IPCC reports make a much wider assessment of the science and methodologies – we do not attempt to reproduce the comprehensive nature of these IPCC assessments here. Our aim is to rigorously

track both climate system change and methodological improvements between IPCC report cycles, thereby transparency and consistency in between successive reports."

We think it is important to track both evolving methodological changes and realworld trends. This way we can achieve the aim of tracking trends between IPCC reports, even if this next report employs very different methods, our paper will still show the trend between the reports. We do not track wider shifts in methodological approaches and developments in the field, leaving these larger assessments to the text IPCC report. We clarify the text in the introduction to further explain our boundaries

**2) FAIR principles**

It is admirable to attempt to follow FAIR (Findable, Accessible, Interoperable, Reusable) principles (Line 111).

In many ways, for the scientific areas examined, the authors have been better at it than the IPCC AR6 report.

However, in some cases I am not sure FAIR principles are being followed as well as they could have been.

We can always to better and strive to improve each year, see responses below.

a) Datasets.

Whilst the providing of data references is excellent, version numbers of source datasets are not always given, and the locations where datasets can be found are not stated.

The supporting information should be the permanent location for such information.

b) Kadow et al. (2020) temperature dataset (that uses HadCRUT5).

I am somewhat perplexed about the continuing use of this dataset.

The publication associated with the dataset used HadCRUT4 as a source.

It is claimed in AR6 and here that this has been updated to use HadCRUT5 as a

source.

As far as I can tell there is no associated peer-reviewed publication for the updated Kadow dataset, and I have been unable to find the resulting dataset online anywhere.

It probably should not have been used in AR6 or in F23, and it should not be used here - unless there is a study describing it and it is publicly available online.

Kadow et al. (2020) analyzed HadCRUT4 because HadCRUT5 was not yet available at that time. However, they did analyze HadSST4, the ocean component of HadCRUT5. The release of HadCRUT5 occurred too close to the AR6 deadline for follow-up methods to be included in time. Kadow et al. (2020) developed an independent method for temperature reconstruction, trained on reanalysis and model data, making it applicable to any temperature anomaly dataset, including HadCRUT5 and just technical adjustments to the data formats. The latter was meant by updated. The transition from HadCRUT4 to HadCRUT5 showed minimal changes in non-filled data, and the trained network used remained unchanged. This means, the reference is still valid. The change to HadCRUT5 is now stated explicitly in the supplement to the paper. An update of the technology and a new publication are in preparation. For this reason, we decided to continue using the published technology to ensure it remains referenceable. See also

Olonscheck, D., Suarez-Gutierrez, L., Milinski, S., Beobide-Arsuaga, G., Baehr, J., Fröb, F., et al. (2023). The New Max Planck Institute Grand Ensemble with CMIP6 forcing and high-frequency model output. Journal of Advances in Modeling Earth Systems, 15, e2023MS003790. https://doi.org/10.1029/2023MS003790

wherein the neural network HadCRUT5 infilled version was analysed.

In future versions we will consider assessing whether there are any further datasets which meet the AR6 criteria. The most likely prospect here is the China-MST dataset, which was used in AR6 and here for land only as its treatment of temperatures over sea ice areas was unsatisfactory. There has been some work on that aspect of China-MST since then and a reassessment may be in order.

c) Result changes between AR6, F23 and F24.

Because of the changes in data, choices about what data to use, and choices in methods, the results will obviously change from year to year somewhat regardless of the direction of any climate change.

The authors do explain this in a number of places (e.g., Line 775) and even show the impact of method/data changes when replicating previous results in a few places (e.g., Figure S2 and table 6).

The impact of method/data changes referred to here, (where last year's results were replicated and compared for the rate of anthropogenic warming), has now also been extended to similarly quantify the breakdown of the increase in the level of anthropogenic warming – see the second last paragraph of section 7.3. The main takeaway from this is that the addition of an *especially* hot/cold year of observations only advances/slows the level of warming by +/- 1 year at current warming rates. This variability is substantially smaller than other contributors to the overall uncertainty and should therefore support our claim that the results are "reliable".

But more context is needed of the impact of data/method changes relative to the given explanations of changes (Table 10).

For instance in Table 3 the effective radiative forcing for the best estimate of total anthropogenic changed from 2.72Wm-2 for 2019 (AR6), to 2.91Wm-2 for 2022 (F23), to 2.79Wm-2 for 2023 (F24) - all relative to 1750.

But it is not explained in the table that the 2019 and 2022 values would be different anyway if replicated with the data used in F24 (Figure 3b).

Comparing the data used in the plotting of Figure 2.10 in Gulev et al. (2021) with the data in Figure 3b in F24 the total anthropogenic ERF is different by up to 0.1Wm-2 throughout the 20th century.

The 2019 value in the F24 data is about 0.08Wm-2 lower than in the AR6 data, a bigger difference than between the AR6 2019 and F24 2023 total anthropogenic values.

This somewhat undermines the reasons given for the differences between 2019 and 2023, and the statement that methodological changes "does not affect historic estimates" in Table 10.

These differences are all small compared to the absolute estimates and their uncertainties, but I only raise the issue given that non-methodological reasons are given for the changes between AR6, F23 and F24 (Tables 3 and 10) and the authors claim the IGCC updates are "reliable" (Lines 79, 944, 955).

We think your point is really only justified for the forcing estimate and the other sections are clear on their methodologies and any changes since AR6. The underlying methodology for assessing the ERF is as close to possible as AR6. The largest change between the 2019 values in AR6 and IGCC2023 is the aerosol forcing. This is due to changes in the CEDS and GFED emissions datasets between AR6 and

IGCC2023 that drive the largest part of short-lived forcer emissions (which in turn are the dominant component of the aerosol forcing). The largest contributors to ERFaci are from SO2 and OC emissions; while SO2 didn't change much from AR6 to IGCC2023 in 2019 (83.7 TgSO2 to 84.2 TgSO2), the OC emissions did (29.8 TgOC to 34.2 TgOC).

The headline assessment for aerosol ERF in AR6 is -1.3 W m-2 for the 2005-2014 average relative to 1750. This has been preserved in IGCC2023. The 1750-2019 assessment of -1.06 W m-2 in AR6 is provided with lower confidence (IPCC AR6 chapter 7, executive summary). It is therefore not unsurprising that the 2019 point value can change with updated emissions that are constrained to pass through a defined point (i.e. 2005-14 of -1.3 W m-2), and one of the reasons why the singleyear estimate in IPCC was given with lower confidence than a multi-year average from a period slightly further back.

For completeness here is the comparison of AR6 with IGCC2023 for 2019 for each anthropogenic component given to three decimal places.

| Forcer               | AR6 2019 | IGCC2023 2019 | Reason for change                                                                                                                                                    |
|----------------------|----------|---------------|----------------------------------------------------------------------------------------------------------------------------------------------------------------------|
| Aerosols             | -1.147   | -1.058        | Updated                                                                                                                                                              |
|                      |          |               | emissions dataset                                                                                                                                                    |
| Contrails            | +0.062   | +0.058        | Change from
approximate
method based on
aviation NOx
emissions in AR6
to the same
method as Lee et
al. (2020) with
newer data in
IGCC2023 |
| Land use             | -0.200   | -0.200        |                                                                                                                                                                      |
| Black carbon on snow | +0.080   | +0.080        |                                                                                                                                                                      |
| CO2                  | +2.158   | +2.156        | Updates to 1750
and 2019
concentrations for
newer X2019
calibration scale                                                                                |
| CH4                  | +0.544   | +0.544        |                                                                                                                                                                      |

| N2O                                                        | +0.208 | +0.208 |                                      |
|------------------------------------------------------------|--------|--------|--------------------------------------|
| Halogenated
GHGs                                        | +0.408 | +0.408 |                                      |
| 03                                                         | +0.484 | +0.474 | Updates to
emission
precursors |
| Stratospheric
waver vapour
from methane
oxidation | +0.050 | +0.050 |                                      |
| Anthropogenic
total                                     | +2.647 | +2.720 |                                      |

Table 10 has been updated to state that the emissions dataset has been updated and materially impacts the estimate of ERF from aerosols (downward revision by 0.09 W m-2 for 2019 relative to 1750).

**3) Detection and Attribution method choices**

I welcome that more explanation is given for the three methods used in the assessment of attributable warming in surface temperatures than was given in AR6 (Eyring et al., 2021).

But some descriptions are a tad over simplistic, and how methods are updated are sometimes not clear.

The three techniques, 'Global warming index' (GWI), 'Kriging for climate change' (KCC) and 'Regularised optimal fingerprinting' (ROF) are all described in the scientific literature.

However, none of the well known limitations of methods such as regression or Bayesian weighting are mentioned in F24, such as how to guard against over-fitting.

Both Ribes et al. (2021) and Gillett et al. (2021) apply imperfect model tests to evaluate the KCC and ROF approaches, including an evaluation of the coverage probability of these approaches (how often pseudo-observations lie within calculated 90% confidence intervals). At least for the analyses presented there, and in the context of the assumption that the real world and climate models are exchangeable, these approaches do not produce over-confident results.

Further, conducting another critical review of the methods that the IPCC authors already reviewed and adopted for AR6 and SR1.5 isn't necessarily within scope these IGCC assessments, with the main outcome being to repeat and update the agreed-upon

assessments, and harmonise approaches where necessary to address potential inconsistencies between IPCC reports.

I think there could be a wasted opportunity to assess whether to continue just using global mean temperatures, and to also use other methods and techniques, and thus move away from the AR6 decisions.

We would suggest that this choice is directly linked to the positioning of this paper. It would be very difficult to reproduce an IPCC-type assessment. Our objective is more modest: to update the calculations without changing the methods. This is why we are using only (and all) AR6 methods, despite their known limitations. Our default approach is to update the AR6 results. It isn't clear which studies/approaches the reviewer is suggesting we should use instead. For information, we did investigate using hemispheric means in Gillett et al. (2021), and it didn't generally result in narrower uncertainties than using global means (and the residual test was failed more often).

Given the relatively low confidence in the results (see (a) below), it may seem pedantic to comment on some of the details of the analysis.

But, I feel it is appropriate given the high precision of the results the authors' chose to use (Line 655) and their interpretation of differences in results between AR6, and F24 (Table 6) - see comment 2c above.

a) 'Likely' versus 'Very Likely'.

It would be helpful to explain why the attribution assessment are given as 'Likely' (Lines 655, 677 and 788).

i.e. "These ranges are then assessed as likely rather than very likely because the studies may underestimate the importance of the structural limitations of climate models... use too simple climate models... or underestimate model uncertainty" in 3.3.1.1.2 in Eyring et al. (2021).

Yes, the choice is exactly as assumed in Community Comment 4 (CC4) – despite the overall multi-method uncertainty range being based on the 5-95th percentile ranges for each of the three attribution methods, AR6 assessed the overall confidence to be *likely* rather than *very likely* (for the reasons quoted from AR6 in CC4). We maintain this conclusion for these repeat assessments. For clarity, we have now explicitly referenced this decision from AR6 in the Supplement Sect. S7.4 where the assessment approaches are detailed.

**b) GWI method**

The GWI method also relies on a simple model - sampling CMIP6 simulated behaviour - and CMIP6 piControl simulations for uncertainties (Line 756 and 764).

The author's claim that the GWI method has a "low dependence on uncertainties in climate sensitivity and forcing" (Supplement Line 305), is over confident.

GWI will be sensitive to evolutionary uncertainties in forcing and in uncertainties in climate feedbacks changes over time.

On the other hand even if correct from the perspective of magnitude of uncertainties, this is not exclusive to GWI - the same statement can be made for most regression approaches, e.g., ROF, as well (Hegerl and Zwiers, 2011), and hints at a limitation of regression methods - the risk of over-fitting.

The difference between the F23 (Supplement Table S3) and F24 (Supplement Table S3) results for 2010-2019 and 2017 (single year) demonstrate that the method is sensitive to end-effects, contrary to what is claimed (Supplement Line 312), but is acknowledged in Line 775 in the main text.

The statement on line 305 was worded a little ambiguously with respect to precisely what it was referring to, and we agree that the statement therefore read as "over confident". This sentence has now been amended to be more precise – i.e. the *central estimate* of human-induced warming from the GWI is very closely tied to observations and therefore this central estimate has a low dependence on the *size* of the uncertainties in climate sensitivity and forcing. The amended statement should no longer be over confident; changing the width of the uncertainties in sensitivity and forcing only minorly affect the central GWI estimate. This property of the GWI was raised in the context of the GWI being regression-based, and was not claimed to be unique to the GWI.

Regarding the end-effect sensitivity, this has now been explicitly quantified for the multi-method assessment in the main text, and also discussed earlier in this response. In addition, the GWI in particular shows a smaller sensitivity to end-effects than the overall multi-method assessment, with the change in the GWI's human-induced warming in 2023 (as assessed in 2023) being 0.03C warmer than in 2022 (as assessed in 2022); this is close to the GWI's rate of warming of 0.024C/year over the last decade; the especially hot 2023 observed end-year therefore only increased the anthropogenic warming by a small additional amount (corresponding to a small fraction of the observed variability last year) compared to what would have been expected from the current rate of warming. Likewise, the differences of 0.01C in the GWI's anthropogenic warming for the 2010-2019 and 2017 periods between the 2022 and 2023 analyses are also small, corresponding (i) (in time) to much less than one year of additional warming at current rates, and (ii) (in magnitude) to only a couple of percent of the observed variability exist in the GWI and are a limitation of the method, but the magnitudes in question are sufficiently small that I would argue that they are consistent with (rather than "contrary to", as suggested in CC4) the

statements made in the Supplement near line 312, of the GWI being (i) "relatively insensitive" to end effects, and (ii) avoiding confusion about a hiatus or acceleration in warming due to large interannual fluctuations in observed warming. c) KCC method

The KCC method is based also on observed warming and energy balance models, as well as CMIP simulations (Line 764).

The explanation of the KCC approach (Supplement S7.2.2) as a Bayesian approach (Supplement Line 346) doesn't really reflect the complex steps involved in the method.

The anthropogenic prior is estimated by smoothing CMIP6 historical simulations, the WMGHG prior from CMIP6 hist-GHG simulations and natural prior from energy balance models tuned to CMIP6 simulations (CMIP5 in Ribes et al., 2021).

This is correct. This information has been added to the supplement (although with slightly less details than described here). We still refer to Ribes et al. (2021) for a full presentation of this method.

I am afraid I am not clever enough to figure out how the posterior distributions were calculated simultaneously for the ANT, GHG, OHF and NAT, so it would be very helpful - not just for myself I am sure - to briefly outline the details in Supplement S7.2.2.

We acknowledge that there is an additional math difficulty to derive the posterior distribution for each term. However, the short S7.2.2 paragraph is only intended to provide a short summary of the method – not full details. Explaining how the posterior is derived requires introducing mathematical notation and duplicating much of the original publication. For this reason, we refer to the Statistical Method of Ribes et al. (2021) for detailed explanation of how KCC can be used for attribution, and in particular, how NAT and GHG induced warming can be inferred, in addition to ALL. Please note that all other forcing combinations (eg ANT, OHF) are derived from ALL, NAT and GHG, assuming linear additivity.

**d) ROF method**

The ROF method is also based on observed warming (Line 766), as well as CMIP simulations.

How much impact did using just three CMIP6 models compared to the 13 models used in AR6 have (Eyring et al., 2021)?

Figure 2e in Gillett et al. (2021) compares the Gillett et al. (2021) 13-model analysis results, with results obtained from CanESM5, MIROC6 and IPSL-CM6A-LR individually (the three models used in this study). The best estimate anthropogenic warming is similar in each case and close to the observed warming, although GHG and aerosol attributable differences are larger. Figure S3 in Forster et al. (2023) shows the net effect of changing the multi-model ROF analysis from 13 models to 3 models, switching from GSAT to GMST, switching from HadCRUT4 to HadCRUT5, and adding an extra 3 years of observations. The mean anthropogenic warming is slightly lower and the uncertainties are reduced, likely largely because the uncertainty in scaling from GMST to GSAT is no longer included. Estimated GHG and other anthropogenic attributable temperature changes exhibit large changes, but overall results are still within the assessed uncertainty ranges from AR6.

The Gillett et al. (2021) study produced an excellent exploration of the sensitivity of the results to different choices, and uncertainty to choice of model. This was also reflected in AR6 by having a figure showing some of those results (Fig 3.7 in Eyring et al. (2021)). It is a pity that those details are not included here.

This is a good idea, we will make such a figure for next time.

e) 'GMST' versus 'GSAT' (Lines 662, Supplement S7.1).

This is all very complex, yet misses some key issues about how reliable it is to estimate 'GMST' from models (e.g., Jones, 2020).

With the published versions of the three attribution methods following different approaches, it was not clear how the AR6 (Eyring et al., 2021) managed to compare the three sets of results in a unified way.

I understand the attempt to present the three analyses in "GMST", but I do not quite see if this is actually what is being shown.

For instance does GWI use 'GMST' or GSAT diagnostics for the CMIP6 models it uses for uncertainty analysis?

It uses a simple model that emulates CMIP6 simulations, were 'GMST' or GSAT diagnostics used in the emulation?

KCC uses HadCRUT5 and following AR6 it assumes that the observed GMST trend is the same as an estimated "GSAT" trend (Supplement: Line 352).

But, what CMIP6 diagnostic was used in the KCC analysis? GSAT or "GMST"?

In Ribes et al. (2021) they use simulated GSAT. How was the attributed GMST estimated?

The KCC analysis shown here still relies on the GSAT warming simulated by CMIP6 models (ie, a global average of tas). Unlike in Ribes et al. (2021), there is no rescaling of historical GMST observations to make them consistent with GSAT – in line with the AR6 assessment. As a result, the KCC estimate can be regarded as a GSAT estimate, consistent with Ribes et al. (2021). Lastly, our estimates (of attributable warming) are considered as valid for GMST because of the AR6 assessment.

For the GWI analysis, the published parameter tunings for FaIR v2.0 are used (from Leach et al. 2021) – these are based on GSAT outputs from CMIP6 simulations, not GMST. A rescaling (as is often done to account for the difference) is not applied to the outputs from FaIR since the rescaling would be immediately regressed away in the next step of the attribution. As noted for KCC above, this approach is also consistent with the AR6 assessment. Only differences in the tuning process that change FaIR's responses beyond a linear rescaling would impact the final attribution results, (and even then only in small way). Exploring this would require a full retuning, which could be done in future, though the differences in the final attribution results would be minor. A comment on this has now been added to the notes on the GWI method in the Supplement for completeness.

4 Statistical interpretation - Aliasing

a) Sort of related to comment 2c above, a little more care should be taken when interpreting changes between overlapping periods.

For instance the difference between observed warming for 2011-2020 and 2014-2023 periods (Table 5), is interpreted as an increase of 0.10C "within three years" (Line 561).

Due to the overlap between the two periods what is actually being compared is Sum(2011-2013 period)/10 and Sum(2021-2023 period)/10, so internal variability over those sub-periods becomes important. Thus 2023 has disproportionately more influence than if an independent 10 year period was being compared with.

This is nodded to in Table 10 "which may in part be internal variability.", but I am not sure the authors are quite aware of the impact of higher frequencies (mostly from interannual internal variability) aliasing onto lower frequencies, when tracking 10 year period results over time.

This is a useful and thoughtful point. We think it is covered off enough within the existing discussion and the additions above.

**Other points:**

A mistake was found in the calculation of the remaining carbon budget, resulting in a slight adjustment of the values in tables 8 and 10. We have also added a few sentences explaining

the budget calculation. There have been other very minor corrections to the data. Therefore new instances, and updated tables and figures have been created. We have also added two authors Alex Borger (alex@cct.earth) and Jiddu A. Broersma (jiddu@cct.earth), as well as an additional figure (new Figure 10) to cover off the visualisation aspects.